

# Detecting and understanding slow glacier flow under climate change: A case study on Vernagtferner, Austria

Theresa Dobler[1], Wilfried Hagg[1], Martin Rückamp[2], Thorsten Seehaus[3], and Christoph Mayer[2]

[1]Munich University of Applied Sciences, Department of Geoinformatics, 80333 Munich, Germany
[2]Bavarian Academy of Sciences and Humanities, Section Geodesy and Glaciology, Munich, Germany
[3]Institute of Geography, Friedrich-Alexander-Universität Erlangen-Nürnberg, Erlangen, Germany

**Correspondence:** Theresa Dobler (theresa.dobler@hm.edu), Christoph Mayer (Christoph.Mayer@lrz.badw-muenchen.de)

**Abstract.** Long-term surface velocity observations of glaciers reflect the dynamics of glacier ice and its interaction with the mass balance, including variations due to climate change. In this study, we investigate the surface velocities of a slow-flowing glacier which is influenced by strong surface melt and negative mass balance during the last decades. The annual stake measurements date back to 1966 and allow the study of ice dynamics for more than five decades. We observed a strong
relationship between the surface velocity and ice thickness, especially in the case of the glacier´s response to thinning. A series of slightly positive mass balances led to a minor glacier advance around 1980, associated with a considerable speed-up of the glacier. With the onset of the negative mass balances, the velocity has decreased steadily until today. Based on recent in-situ measurements, a seasonal variation of surface velocities can be identified, with around 30% higher summer velocities in relation to the annual average. In order to investigate the current ice surface flow, we analyze the potential and limitations
of remote sensing for slow-flowing glaciers. Standard remote sensing techniques did not provide reliable results due to the combination of low ice flow and high ablation, and the associated difficulty in establishing coherence and identifying stable features in the remote sensing products. Instead, manual feature tracking based on a combination of stake measurements and the investigation of unpiloted aerial vehicle (UAV) surveys, and airborne imagery was used to generate a reference dataset for the period 2018–2023. With an average velocity of $1\,\mathrm{m\,yr^{-1}}$ and a maximum displacement rate of $4\,\mathrm{m\,yr^{-1}}$ in the central part
of the glacier, it gives a clear picture of the low present-day glacier flow.

## 1 Introduction

Mountain glaciers move downwards due to internal deformation and, in most cases, basal sliding, induced by the surface slope and gravitational forces. They act as a delayed response to climate change by translating changes in the mass balance into area changes modulated by the ice flow(Heid and Kääb, 2012; Mayer et al., 2013b). The glacier surface velocity, which can be
measured using field or remote sensing techniques, is based on two components: internal deformation of the ice, predominantly due to shear stress and basal sliding at the glacier bed. The glacier surface velocity and its impact on glacier dynamics have been investigated in several studies. For instance, surface velocity changes may provide valuable information on hydrological conditions, including subglacial water pressure and drainage system evolution (Iken, 1981; Mair et al., 2002), on variations in meltwater production (van de Wal et al., 2008) and on changes in glacier geometry and topographic conditions (Heid and





Kääb, 2012; Thomson and Copland, 2017). Thus, glacier velocity serves as a key indicator of multiple processes within the glacier system. Understanding these velocity pattern and their temporal changes on a resolution considerably higher than the spatial variability is increasingly important, particularly for capturing variations in glacier processes. To achieve this, spatial high-resolution velocity information is essential, as it allows a more detailed and precise analysis of glacier behaviour.

There are several methods to measure glacier surface velocities. The first method established by Nye (1959) measures the

displacement of stakes installed at the surface over a period of time to determine the strain rate. This is a very labour-intensive in-situ method, where stakes are placed in boreholes in order to track the movement of the glacier surface. The advantage of this traditional in situ method is the accuracy of displacement measurements over months and years, providing seasonal or annual ground-truth measurements of surface displacement. The disadvantage is the limited spatial coverage due to the point measurement approach, which is further restricted to accessible glacier sections. In addition to traditional methods, modern

remote technologies offer ways to estimate glacier velocities efficiently with large spatial coverage. For instance, data acquired by air- or space-born sensors and UAVs (unpiloted aerial vehicles) can be used to derive glacier surface velocities by using different tracking approaches.

The further development of UAVs offers an economical and simple method to obtain data from a specific area in the desired temporal and spatial resolution. Previous applications are described in detail in Gaffey and Bhardwaj (2020). Despite their

great potential, UAV-based methods also have certain limitations, such as flight time and distance or dependence on weather conditions, especially in the low-cost and open-source software sector, as demonstrated by Groos et al. (2019). Such as UAV measurements, airborne data makes it possible to estimate ice surface velocities over a large spatial area. A variety of sensors, including aerial imagery (Leprince et al., 2008), SAR (synthetic aperture radar) interferometric (Prats et al., 2009) and LiDAR (light detection and ranging) (Arnold et al., 2006) can be applied.

Space-borne remote sensing data can be utilized to estimate ice velocity, particularly in Antarctica, according to Dirscherl et al. (2020). However, these methods have certain limitations, such as temporal and spatial resolution, the need for recognizable features, or maintained coherence across acquisitions. In their investigation across several regions, Millan et al. (2019) showed that sensors such as Sentinel-2 can produce more accurate results than other optical sensors. Global datasets, like FAU's glacierportal (Friedl et al., 2021), `ITS_LIVE` (Gardner et al., 2022) or a dataset published by Millan et al. (2022) offer user-

ready velocity products, providing comprehensive coverage and accessibility for glaciers worldwide. Nevertheless, at very low ice velocities, determining velocity becomes extremely difficult or even impossible. Slow-flowing glaciers need large temporal baselines between image pairs to capture recognizable flow. However, this implies a potential loss of coherence, as surface features (e.g. crevasses) change considerably during this period (van Wyk de Vries and Wickert, 2021).

The aim of our study is to investigate in detail the surface velocities of slow-flowing glaciers, using Vernagtferner as an

example. We analyze the potential and limitations of global remote sensing products and feature tracking with high-resolution imagery for detecting low surface velocities. As previously mentioned, the coherence formation is particularly challenging for very slow-flowing glaciers, such as Vernagtferner, which become more and more typical in the Alps due to thinning as response to global warming. To assess and investigate the limitations of standard remote sensing methods, we have created a reference dataset from a combination of various velocity datasets, derived from manually determined point measurements based on UAV-





surveys, airborne imagery and stake observations. Our focus is primarily on annual velocities, with additional insights into seasonal patterns. Furthermore, we provide long-term, continuous ice velocity measurement information to analyze long-term ice dynamics (from 1966 to present). The resulting reference dataset, combined with the long-term ice dynamics information, supports modeling of alpine glaciers and the evaluation of future remote sensing datasets and methods.

## 2   Study site

The study was carried out on the slow-flowing (Mayer et al., 2013b) alpine glacier Vernagtferner (VF), located in the southern Oetztal Alps, Austria. The glacier covers an area of approximately $6 \, \mathrm{km^2}$ and is still one of the largest glaciers in Austria, with an altitude ranging from around 2900 to about 3500 m a.s.l..VF was selected based on its extensive data availability, suitable for this study. The data archive of VF covers over 130 years of observations, beginning with the first topographic map of the glacier in 1889 generated by Sebastian Finsterwalder (Finsterwalder, 1897). At VF, continuous monitoring of mass balance

and surface velocity started in 1966. Negative mass balance persisted for the last 35 years (WGMS, 2024). The neighboring glaciers Kesselwandferner (KWF) and Hintereisferner (HEF) have also been observed in detail, starting in the 1950s. Together, VF, KWF, and HEF forming a unique monitoring site of glacier-related variables and processes. A detailed overview of the available data sets and measurements is presented in Strasser et al. (2018). VF can be divided into three areas: the western part, called Schwarzwandzunge, which has to be regarded as an independent part since 2006 as a result of the strong glacial

retreat in recent decades, the middle part, known as the Taschach area, and the Brochkogel area in the eastern part (Mayer et al., 2013a). Although Schwarzwandzunge disconnected from the main part, it is still considered as part of VF in order to maintain consistent mass balance measurements of VF over the years. A detailed study (Reinwarth and Escher-vetter, 1999) of the mass balance of the three areas indicates that the Schwarzwandzunge area behaves fundamentally differently from the rest of VF, presumably due to the different area-elevation distribution.



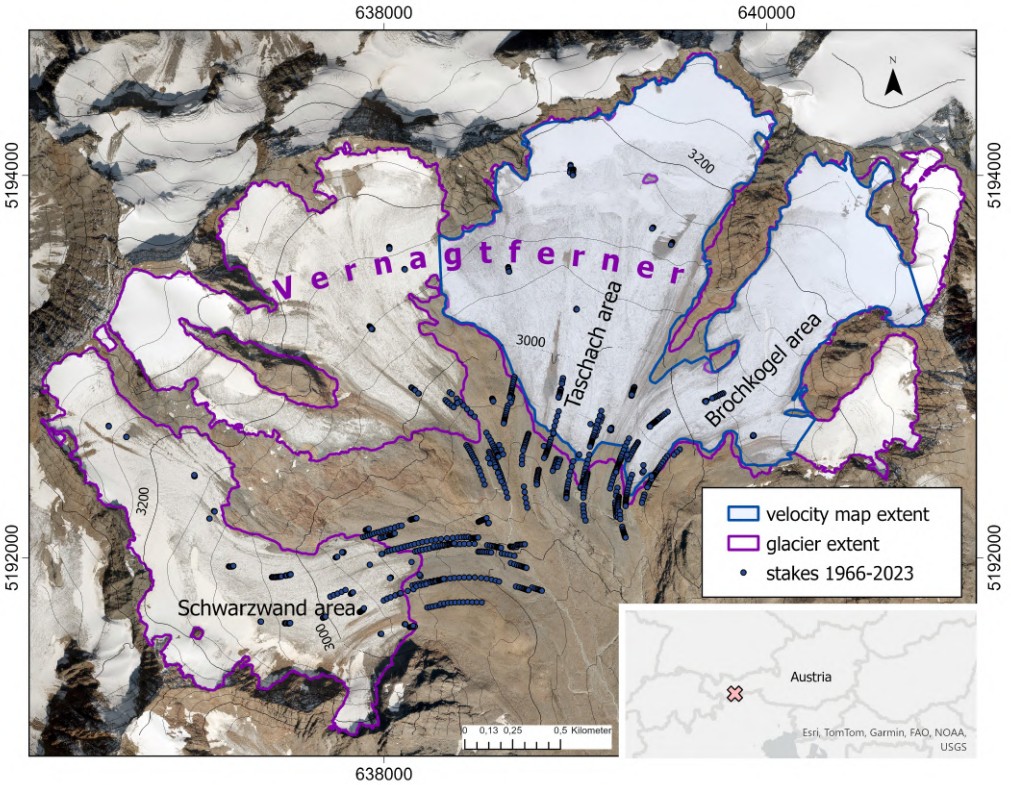

**Figure 1.** Study area Vernagtferner in the Oetztal Alps with the glacier extent from 2016 and the velocity map extent. Background image: Orthoimage © Land Tirol - tiris 2020

## 3  Data and methods

### 3.1  Stake measurements as a long-term data source

A network of stake measurements has been established on VF since 1966. The primary objective of these measurements was the observation of surface ablation and thus the stakes are primarily located in the ablation area (Fig. 1). Therefore, the stake positions are in general not completely representative for capturing ice surface velocities across the entire glacier. The locations of the stakes were measured once a year at the end of the hydrological year (end of September), together with the ablation measurements. This allows the calculation of the annual horizontal displacement and thus the surface velocity.

Over time, different coordinate systems were used as a basis for the measurements. Meanwhile, a local system is used, which is closely aligned with the UTM (Universal Transverse Mercator), and all former measurements have been transformed into this system. Additionally, the methods for determining stake positions changed over time, starting with multiple forward intersections, continuing with terrestrial polar connection, and using GNSS (global navigation satellite system) positioning for



a certain period (Tremel et al., 1994). The derived velocities have been analyzed in parts in previous studies (Hirtlreiter G., 1985; Khazaleh, 2002).

In addition to the continuously monitored stake-network, mainly intended for mass balance measurement, 11 stakes were installed in actively moving sections of the glacier in the Taschach and Brochkogel area during the period 2022–2023. The
stake positions were measured several times (2022-08-04, 2022-07-10, 2022-09-21, and 2023-08-08), allowing an analysis of seasonal variations in ice flow. VF is very likely a temperate glacier, due to its south exposure and strong melting. Therefore, the glacier is not frozen to the bed, which means that a seasonality in velocity is expected due to basal water flow. This is confirmed by water found on the glacier bed throughout a glacier core drilling on VF (Oerter et al., 1982)

## 3.2    Review of existing products

There are user-ready glacier surface velocity products from NASA's `ITS_LIVE` (Gardner et al., 2022), FAU's glacierportal (Friedl et al., 2021) or a dataset published by Millan et al. (2022) for almost all glacier regions worldwide. The databases provide image pair velocity fields for individual glaciers. The results are based on feature tracking algorithms applied to Sentinel-1 Synthetic Aperture Radar (SAR) acquisitions, as well as optical imagery from Sentinel-2 and Landsat missions. The products are based on imagery with spatial resolutions of 10-30 m and provide results at 120-200 m spatial resolution.

At VF the user-ready glacier surface velocities products of `ITS_LIVE`, FAU's glacierportal and Millan´s dataset did not provide usable results. We analyzed the mosaics, which show high spatial coverage and some kind of displacement signal on the glacier surface. However, a closer inspection of the displacement rates and in particular the directions showed that the displacements are very noisy and that the displacement directions are not aligned with glacier flow directions. Figure 2 shows exemplarily the `ITS_LIVE` and Millan data base annual mosaic for 2018, with maximum velocities at the glacier terminus of
the Taschach area.





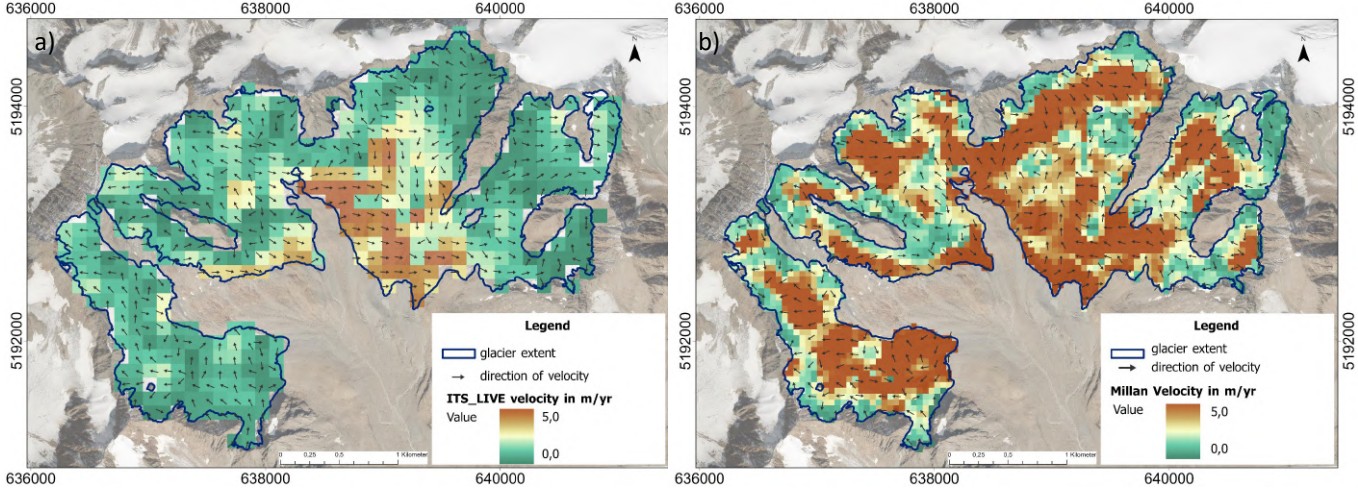

**Figure 2.** Annual surface velocity mosaic from a) ITS_LIVE database and b) Millan database. Background image: Orthoimage © Land Tirol - tiris 2020

## 3.3 Remote sensing measurements

### 3.3.1 Standard satellite remote sensing

Since the user-ready products do not provide usable results, we tested the suitability of high-resolution TerraSAR-X stripmap imagery ( 2 m spatial resolution) for obtaining glacier surface velocity fields. Therefore, we applied feature tracking using
various tracking window sizes and temporal baselines ranging between 11 days and up to about 2 years to pairwise co-registered images. Coherence tracking or InSAR-based displacement measurements were not feasible to carry out at VF, because the InSAR coherence was not maintained between subsequent acquisitions, which we attribute mainly to the pronounced surface-lowering rates in summer and snow accumulation in winter.

### 3.3.2 Manual picking

In order to provide a reliable ground-truth dataset of glacier surface velocities, identical features were tracked manually, using a combination of observations from UAV surveys and airborne imagery. To minimize the impact by potential changes in ice dynamics over the recent years, the analysis was restricted to the period from 2018 through 2023, for which we assume that the ice dynamics did not change much. The data used for this period is summarized in Tab. 1. On VF, crevasses serve as the only reliable surface features for feature tracking, as no other distinctive surface forms are present. In order to reliably find the
same feature in different data sets, only well-identifiable crevasse intersections are used. Because the crevasses are not equally distributed across the glacier, there are areas with spare data coverage.



**Table 1.** Summary of available data showing ice dynamics on VF

| Name of Dataset | Sensor/Instrument | Date of acquisition | Covered area |
|---|---|---|---|
| Airborne 2020 | Optical airborne photogrammetry | 2020-09-08 | Entire glacier |
| Airborne 2018 | Optical airborne photogrammetry | 2018-09-16 | Entire glacier |
| Airborne 2021 | Optical airborne photogrammetry | 2021-09-25 | Entire glacier |
| Airborne 2022 | Optical airborne photogrammetry | 2022-08-23 | Entire glacier |
| UAV 07/2022 | UAV | 2022-07-10 | Parts of Taschach and Brochkogel |
| UAV 09/2022 | UAV | 2022-09-21 | Parts of Taschach and Brochkogel |
| Stake Network | Multiple forward intersection, terrestrial polar connection or GNSS | Annually in September starting in 1966 | Entire glacier, mainly ablation area |
| Stakes at crevasses 07/2022 | GNSS | 2022-07-10 | Parts of Taschach and Brochkogel |
| Stakes at crevasses 08/2022 | GNSS | 2022-08-04 | Parts of Taschach and Brochkogel |
| Stakes at crevasses 09/2022 | GNSS | 2022-09-21 | Parts of Taschach and Brochkogel |
| Stakes at crevasses 08/2023 | GNSS | 2023-08-08 | Parts of Taschach and Brochkogel |

## 3.4 Compiled glacier velocity map

A map of recent velocities on VF has been generated by compiling manual feature tracking of crevasses and stake measurements for the period 2018 to 2023. To achieve a standardized temporal resolution, all measured displacements were converted to a mean annual velocity in meters per year. The measurements taken in summer are corrected for seasonal variations in order to determine a mean annual velocity. Additionally, a special set of features was created to comprise the glacier margin, where a movement of $0\,\mathrm{m\,yr^{-1}}$ was assumed. Towards the margin, ice thickness decreases and consequently glacier dynamics slow down. Although a velocity of close to $0\,\mathrm{m\,yr^{-1}}$ is very likely in these regions, it cannot be directly observed due to the lack of identifiable features. Therefore, specific points were defined to capture the marginal areas adequately.

In order to fill all data gaps and provide results with a uniform spatial resolution, the velocity data were interpolated onto a 50 m x 50 m grid. Given the irregular data distribution and the presence of substantial gaps, natural neighbor interpolation was selected for its suitability in a geophysical context (Sambridge et al., 1995). Natural neighbor interpolation provides an effective balance between computational efficiency and result quality. As a localized method, it ensures that each interpolated point is only dependent on its immediate natural neighbors. A detailed explanation of the method is provided in Amidror (2002). The technique is implemented in MATLAB using the "scatteredInterpolant" function with the "natural" interpolation method.

As previously mentioned, a detailed mass balance study indicates that the Schwarzwandzunge area behaves fundamentally different from the rest of VF (Reinwarth and Escher-vetter, 1999). Due to this differentiation and the scarcity of current velocity data, also from the Hochvernagt area, the compiled velocity map for the period 2018–2023 of this study focuses on the Taschach and Brochkogel areas. The study region for the velocity map is shown in Fig. 1.





## 4   Results

### 4.1   Long term ice dynamic

The examination of historical velocity data is essential to assess long-term trends and investigate the development of glacier dynamics. For this purpose, the data of the stake network, which has been measured annually at VF since 1966, is analyzed. The focus lies on evaluating the temporal evolution of the ice movement. Therefore, only time series of at least six consecutive years are considered and displayed in Fig. 3. The data clearly demonstrate that stake displacements follow the flow structures of the different glacier tributaries. In addition, the generally decreasing distance between the annual stake positions of each data series indicates that stake velocities decrease over time. Figure 4 presents the annual velocity of six particularly long time series, highlighting a brief acceleration phase around 1980, followed by a pronounced and sustained decline. This velocity peak concurs with a period of positive mass balance, while the subsequent decline coincides with a sustained negative mass balance since 1980. This clearly illustrates the sensitivity of the ice movement to changes in surface mass balance.

To evaluate the dependence of the velocity and the mass balance we compared the measured velocities with modeled velocities, derived from local ice thickness and surface slope at the respective time. The calculations are are based on the shallow ice approximation principle according to Hutter (1983). VF is very likely a temperate glacier, and therefore an ice rheology corresponding to $0°C$ is assumed. Since the calculations require mean values of ice thickness and surface slope over distances of the order of several ice thicknesses, the calculations could only be carried out for some suitable stakes. Figure 5 shows a comparison of the measured and modeled velocities for the selected stakes, along with their positions in time. The velocity evolution over the observation period reflects the measurements, with the pronounced velocity peak around 1980 and the subsequent fast decline. The strong sensitivity of velocity to mass balance (shown in Fig. 4) is evident in both the measured and modeled values. While the modeled velocities provide the same dynamic trends as the measured, offsets occur, which can be attributed to uncertainties in the input data (especially the ice thickness), the temporally fixed choice of flow parameters, as well as the unknown contribution of basal sliding. However, it is remarkable that even the small effect of slightly positive mass balance years on the glacier geometry, results in very pronounced changes of ice flow.





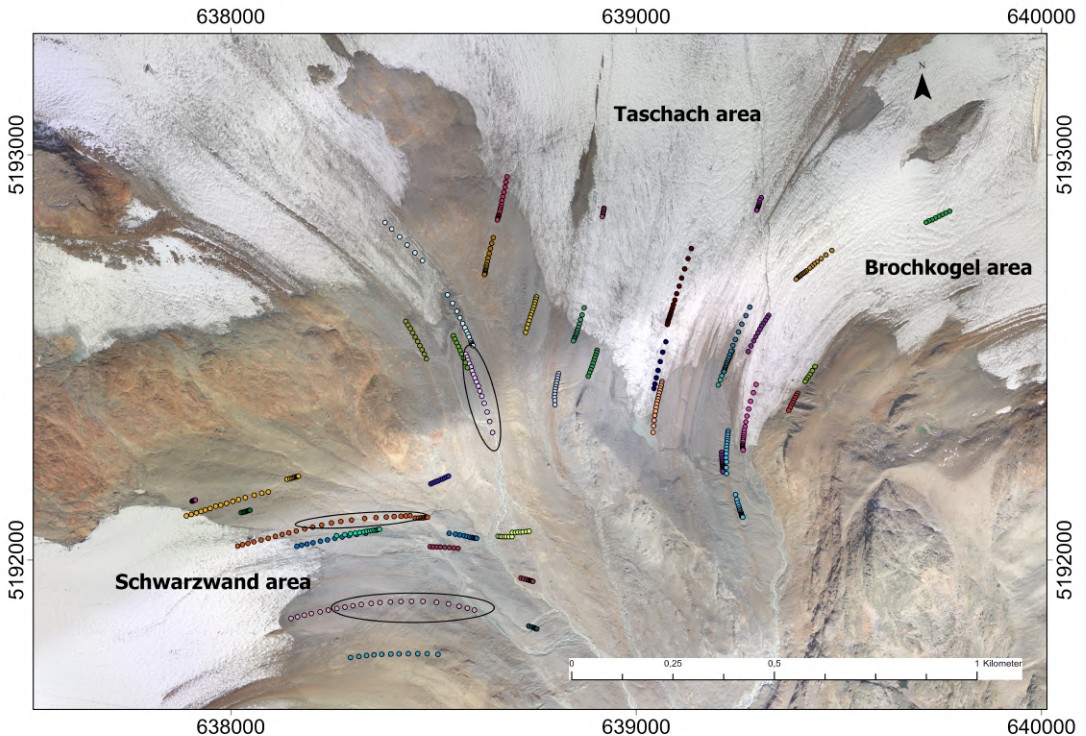

**Figure 3.** Annual position of the stakes measured for at least six subsequent years in the period 1966–2023. The colors indicate the same stake tracked over multiple years. The black ellipses indicate paths that behave differently from the rest, as discussed in Ch. 5. Background image: Orthoimage 2016 © Bavarian Academy of Sciences and Humanities (BAdW), 2016.

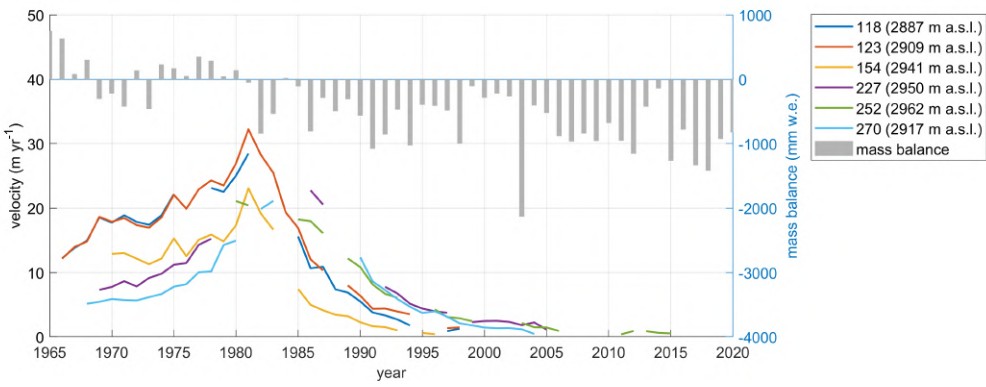

**Figure 4.** Annual stake velocity for particularly long time series and the corresponding annual glacier mass balance. The average elevation of each stake is indicated in brackets to allow an assessment of its respective altitude.



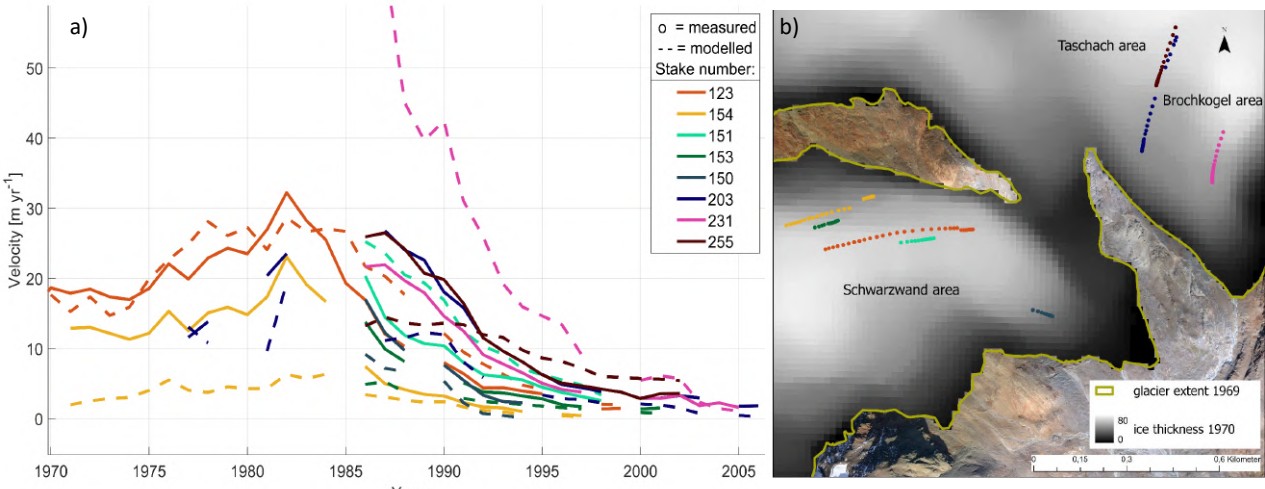

**Figure 5.** a) Measured and modeled stake velocity according to a shallow-ice approximation for selected stakes. b) Position of the selected stakes with the ice thickness from 1970 and the glacier extent from 1969. Background image: Orthoimage 2016 © Bavarian Academy of Sciences and Humanities (BAdW), 2016.

Over time, surface velocities at all altitudes decrease significantly (see Fig. 6). However, for altitudes exceeding 3000 m a.s.l. in the previous years, this trend cannot be conclusively established due to insufficient data availability. In the range of 2900–
3000 m a.s.l. the reduction is most evident. In addition, Fig. 7 shows the regional distribution of surface velocities in connection with the respective ice thicknesses. To provide an ice thickness for each averaged period, the change in ice thickness between the known periods was scaled using the mass balance data at each date. The error of non-linear ice transport over time is neglected, as the velocities are relatively low. Overall, a decrease in velocities can be observed at all altitudes, although their magnitude varies. The decrease in surface velocities is particularly pronounced at the Schwarzwand tongue between the periods a (1970–
1980) and b (1990–2000). In contrast, the decline is less significant in the area of the Taschach tongue. These differences can be attributed to the interaction between ice dynamics and ice thickness. Between the two periods, the ice thickness reduction at Schwarzwand tongue is substantially greater, being approximately 35%, whereas at Taschach tongue, the reduction was only about 10%.



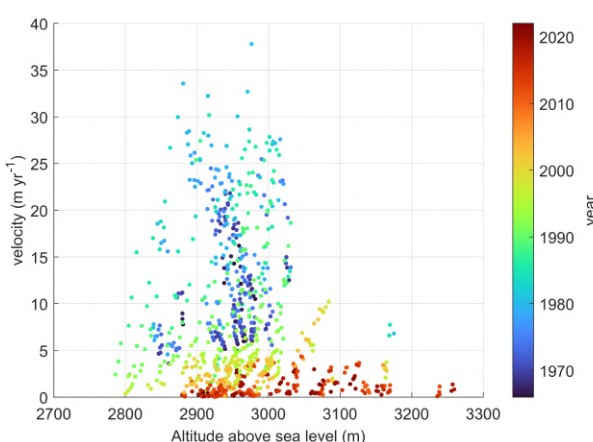

**Figure 6.** Glacier velocities in m yr$^{-1}$ in dependence of altitude and time (color code).

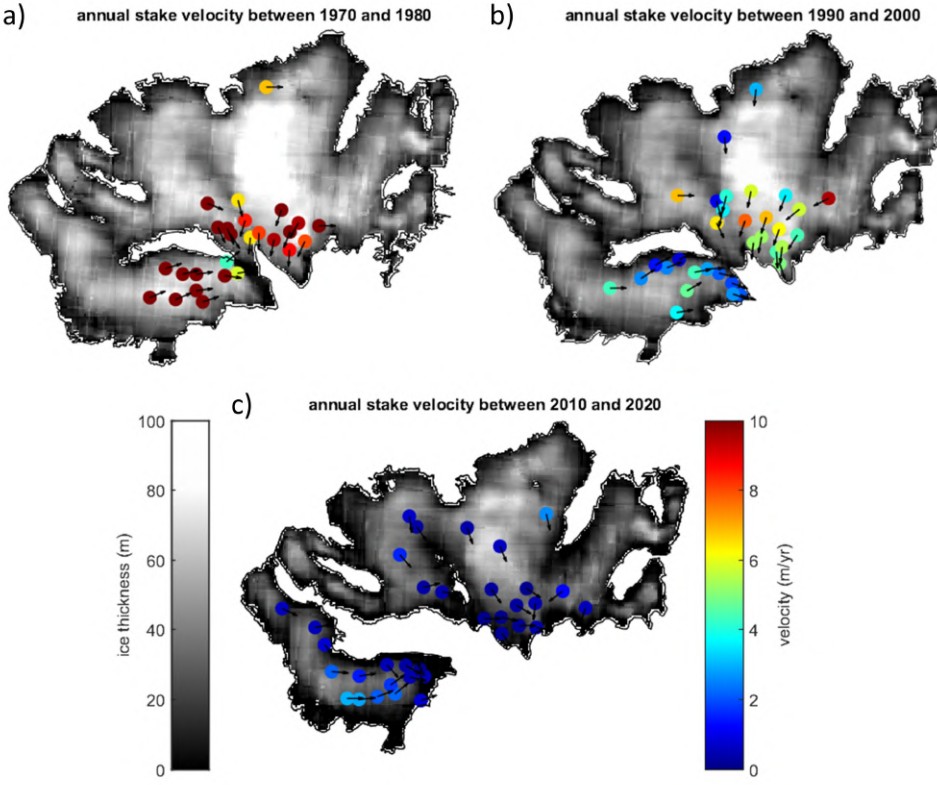

**Figure 7.** Mean annual stake velocities during the indicated period a) between 1970–1980 with 1969 glacier extent and 1975 ice thickness b) between 1990–2000 with 1995 glacier extent and 1995 ice thickness c) between 2010–2020 with 2009 glacier extent and 2015 ice thickness. The arrows indicate the determined flow direction.




## 4.2 Seasonal variation

The measured values at different times (Fig. 8) clearly indicate that ice surface velocities are subject to seasonal variation. This is based on stake measurements in active moving terrain in the period 2022–2023, located in the Brockogel (4 stakes) and Taschach area (8 stakes). In both areas, summer velocities are around 30% higher than the respective annual values. However, we consider the derived seasonality to be representative for the entire VF and assume its applicability to other years. The calculated seasonal variation is used to standardize displacement observations recorded at different times, converting them into

annual velocity values. A seasonal increase of 30% relative to the annual velocity is assumed for the summer months (July through September).

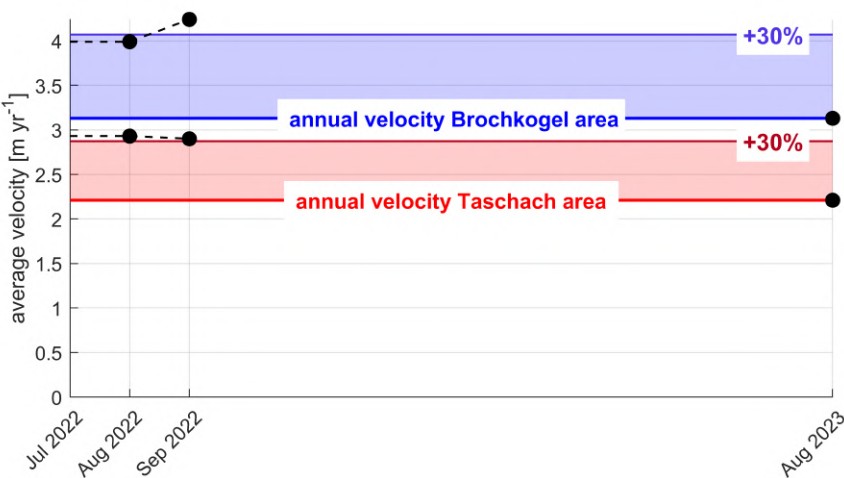

**Figure 8.** Seasonal variation of glacier surface velocity averaged over Taschach (8 stakes) and Brochkogel area (4 stakes).

## 4.3 Present day velocity

### 4.3.1 Standard satellite remote sensing measurements

Since user-ready products do not provide reliable results at VF (see Ch. 3.2), we analysed dedicated image pairs from higher

resolution TerraSAR-X scenes. The microwave scenes also have the advantage of not being influenced by cloud cover. However, this dedicated analysis did not provide satisfactory results. For short temporal baselines (<100 days), the obtained displacement rates are unrealistically high (compared to the ground truth information from stake measurements, see Tab. 2). More realistic displacement rates are obtained for long temporal baselines. However, maximum velocities are located at the glacier tongue, which is also unreliable. Resulting example surface velocity fields from TerraSAR-X data analysis are provided in Fig. 9 for

short and long temporal baselines. Furthermore, no results could be obtained from feature tracking of the high-resolution aerial imagery from subsequent years as the differences between the images did not allow the identification of unique features. Due



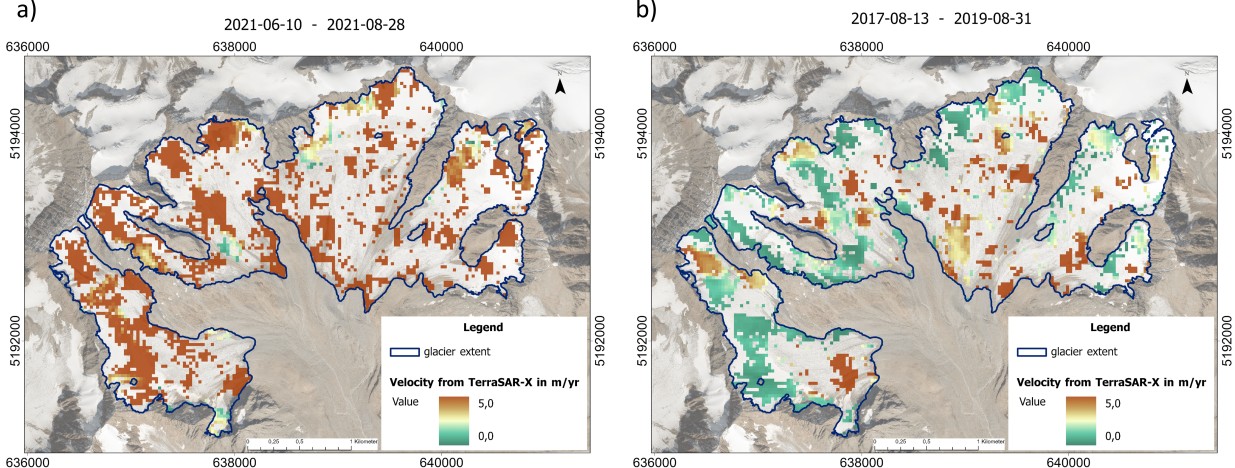

**Figure 9.** Magnitude of the surface velocity field derived from TerraSAR-X images a) from 2021-06-10 and 2021-08-28 (temporal baseline: 83 days) and b) from 2017-08-13 and 2019-08-31 (temporal baseline 748 days).Background image: Orthoimage © Land Tirol - tiris 2020

to inconsistencies in the results from the standard satellite remote sensing data sets with feature tracking, they were excluded from further analysis.

### 4.3.2 Gridded velocity map for a glacier sub-region

It is obviously not possible to (semi-)automatically produce a reliable surface velocity map from aerial or satellite imagery for the slow-flowing Vernagtferner with maximum velocities of less than 5 m yr$^{-1}$. Therefore, we selected a sub-region with a high density of ground measurements for obtaining a gridded representation of ice flow. For this purpose, we combined multiple datasets. Tab. 2 gives an overview of the datasets used and the detected features. However, the data set includes observations from different periods and thus the velocities are affected by seasonal variations. As previously mentioned, on VF it can be

assumed that summer velocities are about 30% higher than the annual average. This seasonal variation was taken into account when converting velocity data recorded at different times into meters per year. For transparency, the calculated correction factor reflecting this seasonal variation is also presented in Tab. 2. A natural neighbor interpolation of all points listed in Tab. 2 was performed. The interpolated velocity map for the period 2018–2023 is presented in Fig. 10. The Figure shows the data points used and the corresponding flow directions. The data points are distributed unevenly, with a higher density in the active moving

zones. The maximum velocities of 4 m yr$^{-1}$ also appear in these active moving zones, while the average surface velocity is about 1 m yr$^{-1}$. In the lower and margin areas, the surface velocity is slightly greater than 0 m yr$^{-1}$. In general, the ice flow direction follows the glacier slope gradient.



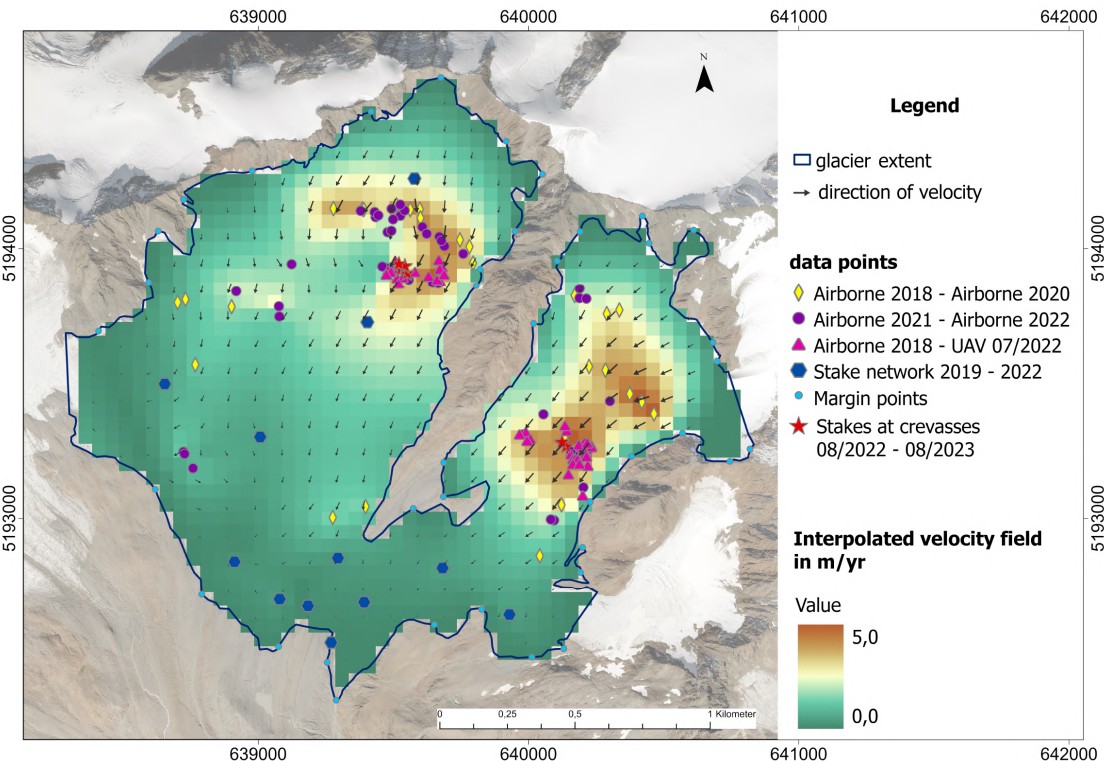

**Figure 10.** Valid map of mean annual velocity for the period 2018–2023 in $[\mathrm{m\,yr^{-1}}]$ with the position of the data points used for the interpolation. Background image: Orthoimage © Land Tirol - tiris 2020

**Table 2.** Compilation of generated velocity points

| Start dataset | End dataset | Generated points | Factor for transformation into $\mathrm{m\,yr^{-1}}$ | average magnitude in $\mathrm{m\,yr^{-1}}$ | Comments |
|---|---|---|---|---|---|
| Airborne 2018 | Airborne 2020 | 31 | /2 | 2.8 | |
| Airborne 2021 | Airborne 2022 | 42 | /0.9 | 2.4 | |
| Airborne 2018 | UAV 07/2022 | 42 | /3.8 | 3.3 | |
| Stake Network 2019-2022 | | 11 | | 0.4 | Small magnitude caused by spatial restriction to tongues |
| Stakes at crevasses 08/2022 | Stakes at crevasses 08/2023 | 11 | | 2.5 | |
| Margin points | | 39 | | 0 | |



## 4.4 Uncertainty Analysis

The accuracy of all stake positions depends on the respective measurement method (multiple forward intersections, terrestrial
polar connection, and GNSS (Global Navigation Satellite System) positioning). In all cases, the placement of a prism or
Antenna, depending on the method, requires a small horizontal offset due to the physical presence of the wooden stakes, which
prevents exact alignment with the center of the borehole. Moreover, if the boreholes are not drilled vertically in the ice, melting
can also cause further displacement biases. In combination with the measurement error, which is in the order of 3 cm, these
factors result in an estimated general uncertainty of $\pm 10$ cm. Therefore, all positions in the stake data set, as well as in the
figures of the multi-year stake displacements, are affected by this uncertainty.

In addition to stake measurements, manually tracked surface features (e.g. crevasse intersections) were used to derive in-
dependent glacier surface velocities. The uncertainty of localizing these features depends on the image resolution (<= 20 cm
pixel size), the feature size (>= 40 cm) and the co-registration accuracy of the images. Considering these parameters, the feature
position uncertainty can be estimated at $\pm 30$ cm.

A covariance propagation law can be used to estimate the uncertainty of the calculated velocities, depending on the temporal
resolution and the data detection method. The resulting velocity uncertainties represent one standard deviation ($1\sigma$) and are
listed in Tab. 3. For instance, feature tracking with a 2-year interval results in a velocity uncertainty of $\pm 0.21$ m yr$^{-1}$, whereas
a 1-year interval yields $\pm 0.42$ m yr$^{-1}$. Since the map of current velocities combines different measurements, a mean overall
uncertainty of all observations can be estimated by averaging the individual uncertainties, weighted by the amount of data
points (see Tab. 3), resulting in an overall uncertainty of $\pm 0.23$ m yr$^{-1}$.

The natural neighbor interpolation of all observations listed in Tab. 3 yields a correlation coefficient (R) of 0.94 for eastward
velocity and 0.95 for northward velocity. The residual variance is 0.15 m, respectively, which indicates a robust and consistent
representation of glacier flow. The reliability of the interpolated map is further supported by the overall alignment of the
calculated flow directions with the glacier topography (slope direction), with no significant outliers observed.

However, the velocity map is least trustworthy in areas with sparse amount of data availability, where interpolation is not
well restricted by actual observations.



**Table 3.** : Statistics of all points used for the interpolated velocity map. The reported uncerainties represent one standard deviation ($1\sigma$).

| Data source | Data detection method | Temporal resolution [yr] | Number of points | Velocity uncertainty [ m yr$^{-1}$] |
|---|---|---|---|---|
| Airborne 2018 - Airborne 2020 | Manual feature tracking | 2 | 31 | $\pm 0.21$ |
| Airborne 2021 - Airborne 2022 | Manual feature tracking | 1 | 42 | $\pm 0.42$ |
| Airborne 2018 - UAV 07/2022 | Manual feature tracking | 4 | 42 | $\pm 0.11$ |
| Stake Network | Stakes | 1 | 12 | $\pm 0.14$ |
| Stakes at crevasses | Stakes | 1 | 11 | $\pm 0.14$ |

## 5 Discussion

### 5.1 Ice dynamics

This subsection discusses the observed changes in glacier flow. Long-term and seasonal variations are interpreted in relation to changes in ice thickness and mass balance. When examining historical data, a general decrease in glacier velocity over time is prominent. This trend can be attributed to two main factors. First, the stakes slow down as they approach the glacier tongue, in line with the general velocity decrease towards the tongue. Second, the reduction in ice thickness due to the continuous negative glacier mass balances influences the velocities, as deformation rates are directly dependent on ice thickness. On VF the magnitude of velocity change varies depending on the region. This can be seen particularly at Schwarzwand tongue, where both the ice thickness and the velocities decrease significantly, even more than on the rest of VF. As ice thickness decreases, deformation rates and velocities decline, a well-documented pattern in glaciological studies. One exception to the general slowdown is evident, as marked by three black circles in Fig. 3. These cases show a temporary increase in velocity during the period 1975–1985. The accelerated ice flow during this period likely originated from the temporarily positive mass balances and an accompanying small glacier advance around 1980 (Mayer et al., 2013b). Neighboring glaciers, such as Hintereisferner show a similar velocity peak around 1980 (Stocker-Waldhuber et al., 2019). This observation aligns with a wider regional trend, as a significant proportion of glaciers in the European Alps experienced advances during this temporarily relatively cold phase (Patzelt, 1985; Wood, 1988).

Short-term seasonal variations play a role in the general glacier flow behavior. A significant temporal variability is observed, with a wide range of notable differences between the summer and winter seasons. There is general agreement that this variability is influenced by the glacier size, as well as internal and subglacial hydrology, although this has not been clarified conclusively and is still a subject of current research (Vincent and Moreau, 2016; Sanders et al., 2018; Nanni et al., 2023; Troilo et al., 2024). In general, the seasonal variability can be rather large, with summer velocities either comparable or even up to 300% higher than winter values (Nanni et al., 2023; Troilo et al., 2024). At Vernagtferner, summer velocities are approximately 30% higher than the annual mean. Our measurements were conducted in summer 2022 in an area of higher ice flow, which is relatively high-elevated, but it is still not part of the accumulation zone and is strongly affected by melting. Meltwater input





and the related increase in subglacial water pressure can reinforce the basal sliding of the glacier during the summer until an efficient drainage system is developed (Iken and Bindschadler, 1986). Crevasses and moulins can encourage the meltwater input to the glacier bed, increasing the basal sliding (Das et al., 2008). At our measurement area, such an increase in meltwater input can occur by existing crevasses, potentially also from upstream regions. As a result, seasonal variation may be stronger in

our measurement area, particularly as efficient drainage systems are probably only well developed further down on the glacier tongue. Whether this summer velocity increase is representative for larger areas of VF and for other years has yet to be demonstrated. No assessment can be made regarding the time of highest seasonality because of the lack of measurement data, other studies refer to spring (Iken and Bindschadler, 1986; Bartholomaus et al., 2008), due to high melting rates combined with a not yet developed drainage system.

## 5.2 Implications for observations of glacier velocities


This part outlines the capability and limitations of current remote-sensing methods in capturing glacier flow, on a slow moving glacier. The annual surface velocities at VF are on average around $1\,\mathrm{m\,yr^{-1}}$ while maximum velocities reach around $4\,\mathrm{m\,yr^{-1}}$. These low velocities pose a critical issue for measuring the glacier flow using remote-sensing techniques like feature tracking. A sufficient displacement of the features between the consecutive acquisitions is needed. The retreat dataset is based on consec-

utive Sentinel-1 SAR acquisitions with minimum temporal baselines ranging between 6-48 days, depending on the region, and a maximum of 96 days (Friedl et al., 2021). For the Alps, a minimum temporal baseline of 48 days was employed. However, considering a spatial resolution of Sentinel-1 IW data of 10 m, the displacement rate at VF would correspond to 0.025 pixels for 96 days between the data takes, which is too small to be accurately captured by feature tracking. Typically, upsampling rates of 2-8 are employed to obtain sub-pixel displacement rates. However, the displacement rates at VF are too low, even

for high-resolution TerraSAR-X stripmap data (2 m resolution). Moreover, the accumulation area has decreased significantly in recent decades, leading to widespread surface melting across large parts of the glacier (WGMS, 2024). The magnitude of point mass balance due to melting exceeds horizontal velocities in many areas on VF, meaning that observed feature velocity is heavily overlaid by surface changes caused by melting in summers, further limiting the unambiguous identification of features between the consecutive acquisitions. Interferometric techniques, such as SAR coherence tracking or InSAR displacement

measurements, are hampered as well by the strong ablation rates during summers and snowfall throughout winters. These climatic factors also explain the observed limitations in the tacking results when using long temporal baselines, as tested for the TerraSAR-X data and selected `ITS_LIVE` image pair velocity fields. Moreover, the limitations in the individual image pair results can not be compensated by combining multiple results in monthly or yearly mosaics, since all output is affected by the various issues (see Figs. 2 and 9 and Section 4.3.1). Overall, the limited displacement rates, high surface melt during summers

plus high accumulation during winter as well as the irregular distribution of surface features impede the capturing of the ice flow at VF by standard remote-sensing techniques. This illustrates the limited applicability of current remote sensing methods for measuring surface flow in slow-flowing glaciers.

Our work reveals that standard remote-sensing velocity products (this study and other products) have limitations or even fail to retrieve the ice surface velocities of slow-flowing glaciers. Due to these limitations, ground-truth measurements and/or





manual feature tracking are still the method of choice, although the methods are very demanding compared to standard remote sensing techniques. The manual feature tracking enables a focused segmentation of high-quality points, a task that remains challenging even to the human eye due to the significant surface changes over time. It allows simultaneous plausibility checks of individual points, ensuring greater reliability.

Although we were not able to generate a surface velocity map for the entire VF, we successfully constructed a valid map

of current velocity of the Taschach and Brochkogel area, demonstrating key dynamical mechanisms operating on the glacier. The map integrates multiple datasets for the period 2018–2023, primarily stake measurements and manually tracked crevasse features. Consequently, areas with many crevasses are represented by numerous data points, whereas regions lacking distinct surface features are sparsely covered, where stake measurements remain the only basis for the interpolated velocity map. A clear correlation is evident between the appearance of crevasses, indicated by areas with a high density of data points, and high

velocities, up to $4\,\mathrm{m\,yr^{-1}}$. A connection of crevasses and velocities is well-documented in the literature, including detailed studies such as Vaughan (1993).

## 6   Conclusions

The ice motion of slow-flowing glaciers and the challenges in detecting it are presented in this study using VF as an example. Long-term ice dynamic observations, recorded since 1966, reveal insights into the evolution of glacier dynamics under

changing climatic conditions. A minor glacier advance around 1980 corresponds to a time of positive mass balance. As a consequence, there was a noticeable increase in velocity for a short period. A long series of negative mass balances during the subsequent years resulted in a significant decline in the surface velocities. Ground-truth measurements enable the estimation of seasonal variations in ice flow, with summer velocities being up to 30% higher than the annual average. The estimation of surface velocities turned out to be challenging for slow-moving glaciers. Standard remote sensing methods were unsuitable

for determining velocity, due to the low flow velocities and strong surface melting, which rendered automated feature tracking and coherence establishment unreliable. To obtain a reference dataset for the years 2018-2023, a velocity map was created manually using a combination of stake measurements and feature identification from UAV surveys and airborne imagery. With the general tendency of strong negative mass balances and the related slowdown, it will become increasingly difficult to determine surface velocities and thus ice flow on numerous glaciers across the Alps. Here, we highlight the challenges in deriving

velocity information for such glaciers and demonstrate the value of long-term observations for investigating the change in ice dynamics due to continuous reduction in ice thickness. The results are a useful starting point for understanding the behaviour of slow-flowing alpine glaciers and their variability in response to changing environmental conditions. The generated velocity map (available at Dobler et al. (2025)) allows a detailed modeling of glaciological processes as well as the evaluation of future remote sensing datasets, but is only valid for the observed period (2018–2023), with the highest uncertainty in the regions

with low data coverage. Overall, the mean uncertainty of all observations can be estimated at $\pm 0.23\,\mathrm{m\,yr^{-1}}$, with no outliers included. This indicates a high degree of reliability and thus provides a solid reflection of the glacier flow. However, a contin-



uation of long-term in-situ observation is essential for a better understanding of glacier dynamics and their response to climate change.

*Data availability.* All data described in this study are documented and will be available on PANGAEA. During the review process, the data
is temporarily available from URL: https://syncandshare.lrz.de/getlink/fiVSQJDUox3YZi87kGD74Y/ in the folder Data under a CC-BY 4.0 license. (Dobler et al., 2025)

The first dataset (displacement_2018_2023_.txt) describes the displacement of glacier surface points (e.g. from UAV, airborne photogrammetry) in the period 2018-2023. The second (stake_network_1966_2023_.txt) describes the stake displacement of the installed stake network in the period 1966-2023. The third (velocity_map.nc) describes an interpolated map of current velocity, based on displacement information
(stake_network_1966_2023_.txt and displacement_2018_2023_.txt), using only data from the 2018-2023 period. There is also a data set (seasonal_variation_data.txt) that describes stake measurements at different times. These measurements can be used to estimate the seasonal variation. In addition, there is a read_me.txt, which includes information of the datasets.

*Author contributions.* TD compiled the data, prepared the figures, conducted most of the processing and analysis (excluding standard satellite remote sensing components), and wrote the majority of the manuscript. CM accounts for the maintenance of the long-term monitoring
programs, provided conceptual input to the study design, and expertise for the evaluation of the results, particularly concerning historical data trends. TS performed the analysis of ITS_LIVE and TerraSAR-X velocity data and contributed particularly to the development of the standard satellite remote sensing section of the manuscript. MR offered ideas for structuring the workflow and contributed to the evaluation and discussion of the results. WH provided detailed comments and corrections on the manuscript. All authors supported the analysis and commented on the manuscript.

*Competing interests.* The authors declare that they have no conflict of interest.

*Acknowledgements.* We would like to thank all the people who have carried out the extensive fieldwork over the years, making this comprehensive series of measurements feasible. The long-term observation at Vernagtferner was made possible by the Bavarian Academy of Sciences and Humanities. This work was financially supported by the Munich University of Applied Sciences HM and the Deutsche Forschungsgemeinschaft (DFG, German Research Foundation) – Projectnumber 512819356. TS was financially supported by the Emmy
Noether Programme of the Deutsche Forschungsgemeinschaft (Grant No. SE3091/5-1)



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
