# Peer review of "Detecting and understanding slow glacier flow under climate change: A case study on Vernagtferner, Austria"

_EGUsphere, 2025_

## Author Comment (AC1)

**Author´s response to referee comment # 1**

First of all, we would like to thank the reviewer for such a detailed and constructive review. It has shown us which aspects need to be explained in more detail and has certainly helped us to improve the manuscript.

In the following, we quote the reviewer´s comments followed by our replies, which are marked in orange.

**General comments**

The manuscript reports on the temporal evolution of surface velocity of the glacier Vernagtferner in the Eastern Alps with focus on a period of major slowdown in recent years after the glacier broke up in several parts. The presented velocity data are based on in situ stake measurements and manual feature tracking in optical airborne images. The presented material and related discussion cover two main topics: (i) the presentation and description of the derived velocity field and of factors responsible for slow-down, including a review of the glacier behaviour since 1966; (ii) an overview on specific remote sensing products on glacier velocity, leading to the conclusion that these products are not suitable for application to slowly moving glaciers. Topic (i) is an interesting case study on a glacier in retreat, demonstrating the impact of long-term negative mass balance and related glacier thinning on a previously quite active Alpine glacier. Factors responsible for the decline are discussed, but an in depth assessment of the governing processes is missing. The presented material on topic (ii) does not contain any novel aspects. In summary, the paper presents an interesting case study on slow-down of an Alpine glacier in decay but has substantial shortcomings. Major revisions and shortening (in particular regarding topic ii) are required.

Topic (i): We have included a classification of the relevant processes for slowing down glacier velocity. Please see in the last comment of this response.

Topic (ii): We agree with the reviewer, that this topic does not provide any new aspects. However, the use of global data sets and standard remote sensing would be the most obvious choice, as these data also provide velocity maps for the area. We aimed to demonstrate that neither the existing global datasets nor our remote-sensing attempts specifically focussed on VF provide/are able to generate a reliable velocity field. We do not want to rule out the possibilities of semi-automated feature tracking. Rather, we would like to explain why we chose manual feature tracking and point out the problems caused by ablation in more detail (see comments). Based on this finding, we generate an interpolated velocity map based on in-situ measurements. The focus of the manuscript is more on the historical data sets, whereby we believe that an examination of current ice dynamics is relevant in order to provide a velocity map ready for other studies, e.g. ice flow modelling. We will try to better clarify this in an updated manuscript.

**Specific Comments**

**Satellite remote sensing:** Major parts of the manuscript refer to spaceborne remote sensing of glacier velocities (Introduction line 45 to 53, Sections 3.2, 3.3.1, 4.3.1, Section 5.2, Fig.2, Fig. 9). These sections refer to properties and suitability of particular products, but do not provide any novel information on methods, accuracy assessment and constraints regarding remote sensing techniques and products for mapping glacier surface motion. During more than three decades results of detailed performance analyses on glacier surface velocity methods and products have been reported in publications and product specification documents, both for so-called "standard satellite remote sensing products" as well as for products derived from data of different satellite missions. These publications show that the pixel size and errors of the "standard products" (based on offset tracking) do not match the accuracy and spatial detail required for mapping very slow velocities at comparatively small spatial scale as observed on Vernagtferner. Taking this into account, the sections on satellite remote sensing can be largely shortened and replaced by references on documents and publications specifying performance numbers for specific satellite-based velocity products. Furthermore, Figures 2 and 9 can be omitted because there is no need showing examples of deficient velocity maps using input data that are not matching the technical requirements needed for velocity retrievals of the study glacier.

Thank you very much for your feedback. You are absolutely right, further information on the accuracy assessments is missing, especially for the velocity products. We have added some information on this. We also agree that, due to the pixel size and errors, the "standard products" cannot provide any meaningful values for a slow-flowing glacier such as the VF. However, we would like to point out that many "standard products" still provide values for the region, even though, as you have already mentioned, it is clear that these cannot be meaningful. Slow-flowing glaciers (which make up a large proportion of Alpine glaciers) should therefore be excluded from the complete "standard products."
We have now explicitly included this note.
We do not want to rule out the possibility that, in principle, no velocity data can be derived for the VF via remote sensing. Therefore, in Section 3.1 with Fig. 9, we would like to include an analysis of the significantly higher-resolution TerraSAR-X data with a spatial resolution of 2 m, which makes detection quite realistic, assuming a detection capability of 0.1 of the pixel size. On VF the challenge is not only in the slow flow, but also in the high ablation and therefore the change of features due to ablation. To explain this in more detail we added an Appendix.

*Text:*
*Chapter: Review of existing products*
*There are user-ready glacier surface velocity products from NASA's |ITS_LIVE| (Gardner.2022), FAU's glacierportal (Friedl.2021) or a dataset published by (Millan.2022) for almost all glacier regions worldwide. The databases provide image pair velocity fields for individual glaciers. The results are based on feature tracking algorithms applied to Sentinel-1 Synthetic Aperture Radar (SAR) acquisitions, as well as optical imagery from Sentinel-2 and Landsat missions.*

The products are based on imagery with spatial resolutions of 10-30 m and provide results at 120-200 m spatial resolution. *The accuracy of the products can vary strongly locally and depends on the velocity level, coherence, existing features, etc., but can be estimated at 0.08 m per day for a temporal baseline of 12 days, for example (Gardner.2022,Friedl.2021,Millan.2022). The accuracy is insufficient for a slow-moving glacier such as the VF, as the speed is lower than the uncertainty of the measurement. Nevertheless, the products still provide values for this area.* We analyzed the mosaics, which show high spatial coverage and some kind of displacement signal on the glacier surface. However, a closer inspection of the displacement rates and in particular the directions showed that the displacements are very noisy and that the displacement directions are not aligned with glacier flow directions. Figure 2 shows exemplarily the ITS_LIVE and Millan data base annual mosaic for 2018, with maximum velocities at the glacier terminus of the Taschach area.

*Appendix A:*

*The figures show the same situation at different points in time:*

[Figure]

UAV- 2022-07-10                    UAV- 2022-09-21

Legend
● Stakes 2022-07-10
● Stakes 2022-09-21

*Fig. 2:*. Melt-induced changes to crevasses. *The example is taken from the upper part of the Taschach area at an altitude of approx. 3150 m. Maximum velocities occur in this area at the VF.*

*Figure 2 clearly shows a change in surface features, in particular a widening of the crevasses. We examined two of the crevasses in more detail using two stakes at the upper and lower edges of each crevasse. The average movement of the eight stakes (shown in Fig. 2) over the two months is approximately 0.73 m (change from the red to the blue points), with a standard deviation across the eight stakes of 0.12 m . The stakes rule out the possibility that a significant actual break-up of the crevasses has taken place. The stakes show an even shift of the upper and lower edges of the crevasses. Thus, the change in the surface is probably largely due to melting. Even if identical features (in this case, crevasse edges) could be identified during the period, the movement would be significantly overlaid by ablation due to the crevasse edges (and their varying melt rates), resulting in erroneous dynamics.*

*Even with larger temporal baselines, manual feature tracking can be used to exclude features that are likely to have changed due to ablation, as is often the case in 2022. Crevasse trace intersections are particularly well suited for this purpose, as can be seen in Fig. 3 Identifying identical features that have not been subject to high ablation is challenging even for the human eye, especially over longer baselines, such as the example in Figure 4 over a period of two years.*

[Figure]

*Fig. 3:* Melt-induced changes to crevasses. *Example from the upper part of the Brochkogel area at an altitude of about 3200m.*

[Figure]

*Fig. 4:* Crevasse pattern in a period without significant melt. Crevasse pattern in a period without significant melt. *Example from the highest part of the Brochkogel area at an altitude of about 3250 m.*

**Analysis and interpretation of surface velocities:** The new velocity data, presented in the manuscript, refer to the period 2018 to 2023 when maximum velocities of Vernagtferner were below 5 m yr$^{-1}$. The presented velocities are horizontal displacements referring to individual points on a main branch of the glacier, based on manual feature tracking in optical airborne images and on stake measurements. Considering the average velocity of 1 m yr$^{-1}$, it is obvious that the magnitude of vertical surface lowering exceeds that of the horizontal

displacements. Consequently, the vertical displacement of the surface (mass depletion due to surface/atmosphere exchange processes) is the dominating component for the glacier mass balance in the current state and ice dynamics plays a minor role. In this context, quantitative information on the annual mass balance (respectively the related topographic change) and its spatial pattern during the study period would be of interest.

We agree. We provided further analysis. Please see the last comment of this response.

Line 45 to 53: This is a one-sided introduction on space-borne remote sensing applications for ice velocity monitoring. The statement on the use of space-borne remote sensing "particularly in Antarctica" does not reflect the actual situation in which ice velocity products are generated routinely on behalf of various organizations, covering at large all global land ice areas. Several of these products exploit also radar repeat-pass interferometry in regions and seasons where coherence is preserved.

That´s true! The specialization in Antarctica should refer specifically to the cited literature. To avoid misunderstandings, we will rephrase this:

Space-borne remote sensing data can be utilized to estimate ice velocity of mountain glaciers and in particular of the large polar ice sheets (Dirscherl et al., 2020).

Line 56-57: Temporal decorrelation is not a particular problem for slowly flowing glaciers, but rather for fast movement, particularly in shear margins where interferometric fringes are often tightly spaced or aliased.

That is correct. It is the combination of high ablation (and the associated significant surface change) and low velocities that makes coherence formation difficult, as the very low velocity is barely measurable when coherence exists or the feature change is overlaid by ablation instead of actual ice dynamics. Therefore we changed the sentence to:

 As previously mentioned, the combination of high ablation (and the associated significant surface changes) and low velocities leads to challenging remote sensing detection of velocities. When coherence exists, the very low velocity is hardly measurable and most likely overlaid by feature changes due to melting. This combination of high ablation and slow flow is becoming increasingly  typical in the Alps due to thinning in response to global warming.

Line 86: Horizontal displacement and surface velocity are not the same.

Thanks for mentioning that. We change it to:
This allows the calculation of the annual surface velocity.

Line 113-118: Whereas time spans of TerraSAR-X repeat-pass data are 11 days, there are other high resolution SAR constellations offering shorter repeat-pass sequences, well suitable for glacier velocity mapping based on the motion-related interferometric phase. For example, successful glacier monitoring applications have been reported for 1-day, 3-day and 4-day interferometric repeat pass pairs of the COSMO Sky-Med constellation, providing high accuracy velocity products. Coherence tracking applies cross-correlation matching of templates and thus has a spatial resolution and sensitivity similar to feature tracking.

You are absolutely right; in this paragraph, we describe the TerraSAR-X data in a very one-sided manner. We have added information about possible further (including shorter baseline) satellite missions, which have already been used to generate very good velocity information for faster-flowing glaciers. Since the average velocity on the VF is only about 1 m/year, which corresponds to an average movement of about 0.27 cm per day without taking possible seasonal fluctuations into account, baselines of 1-4 days for interferometric displacement mapping, might lead to some results. However, there are no acquisitions available for VF. Moreover, the considerable surface melt in summer will limit this application to winter months, where snowfall also leads to the loss of coherence.

*We change the manuscript to:*
In addition to user-ready products (which do not provide usable results), there are high-resolution SAR constellations offering repeat-pass sequences beginning at 1-day. To name a few, ICEYE InSAR (Lukosz 2021), Capella Space SAR (Izzard 2025), COSMO Sky Med (Wang 2018), and TerraSAR-X (Schubert 2013) constellations have already been successfully used to generate velocity maps. As an example, we tested the suitability of TerraSAR-X stripmap imagery (2 m spatial resolution) for obtaining glacier surface velocity fields, on VF. Therefore, we applied feature tracking using various tracking window sizes and temporal baselines ranging between 11 days and up to about 2 years to pairwise co-registered images. Coherence tracking or InSAR-based displacement measurements were not feasible to carry out at VF, because the InSAR coherence was not maintained between subsequent acquisitions, which we attribute mainly to the pronounced surface-lowering rates in summer and snow accumulation in winter.

Figure 3: Please provide information on the time periods (years) to which the time sequences of the individual stakes refer.

Due to limited display and space constraints, it is difficult to provide precise information on the exact period of time in this figure. The periods of the longer (more relevant) time series can be read from Figures 4 and 5, as the same color coding has been used here. We have added a note to the caption of Figure 3 to indicate the color correlation with the following figures:
The colors correlate with Figures 4 and 5, which show the temporal resolution of most stakes.

Figure 5 caption: Please provide reference for the source of the ice thickness data 1970.

Thank you for the hint. We have added it, the new caption for Figure 5 is now:

Figure 5. a) Measured and modeled stake velocity according to a shallow-ice approximation for selected stakes. b) Position of the selected measuring stakes with the glacier extent from 1969 and the ice thickness from 1970 (calculated from the bed topography from 2006 (

Mayer et al 13b) and the glacier surface at known points in time (Finsterwalder 1972, Rentsch 1982), scaled using the mass balance), interpolated using the mass balance. Background image: Orthoimage 2016 © Bavarian Academy of Sciences and Humanities (BAdW), 2016.

Furthermore, this information was also added to the Figure 7 caption:
 The respective ice thickness is calculated from the bed topography of 2007 (Mayer et al., 2013b) and the glacier surface at known times (Finsterwalder, 1972; Rentsch, 1982 and more recent maps published by the Commission for
Geodesy and Glaciology), temporally interpolated by the mass balance.

Section 4.2, Seasonal variation: The summer velocities are based on measurements in summer 2022, the ablation period of the mass balance year 2021/22 with the largest mass deficit of the 2018 to 2023 period in the Eastern Alps. Consequently, it is unclear if the number for the seasonal velocity increase 2022 is representative for the whole period.

Thank you for pointing this out. We have added the reference to the extremely negative year.

*Text:*
*However, we consider the derived seasonality to be representative for the entire VF and assume its applicability to other years, although 2021/22 was a year with an extremely negative mass balance.*
*The calculated seasonal variation is used to standardize displacement observations recorded at different times, converting them into annual velocity values. A seasonal increase of 30\% relative to the annual velocity is assumed for the summer months (July through September). Thus, the seasonality measured in 2022/23 is also assumed for the years 2018--2023.*

Figure 8: The presentation of a few numbers (velocities, exact dates) in the form of a table would be more appropriate than the display within a diagram.

Precise data (not averaged) is certainly interesting at this point, which is why we have included a table now. However, to enable quick and easy reading of the 30% seasonal variation, the table has been added in addition to the figure.
The coordinates of the individual measurements are stored in the PANGAEA Database. The table only shows the resulting velocities, separated into the Taschach and Brochkogel areas, whereby the Brochkogel area is only represented by 3 respectively 4 stakes within a few square meters, meaning that they all describe the same situation  (one point could no longer be found in August 2023).

**Table 2.** Exact values of the seasonal variation of glacier surface velocity for Taschach (8 stakes) and Brochkogel area (4 stakes).

| Area | Number | $v_{Jul-Aug}$ Juli 22–Aug 22 [m yr$^{-1}$] | $v_{Aug-Sep}$ Aug 22–Sep 22 [m yr$^{-1}$] | $v_{annual}$ Aug 22–Aug23 [m yr$^{-1}$] | $\frac{v_{Jul-Aug}}{v_{annual}}$ seas. var. | $\frac{v_{Aug-Sep}}{v_{annual}}$ seas. var. |
|---|---|---|---|---|---|---|
| Brochkogel | 1 | 3.37 | 5.05 | 3.14 | 1.07 | 1.61 |
| Brochkogel | 2 | 3.44 | 4.74 | 3.03 | 1.14 | 1.56 |
| Brochkogel | 3 | 4.52 | 4.16 | 3.22 | 1.40 | 1.29 |
| Brochkogel | 4 | 4.62 | 3.00 | no measurement | | |
| **Brochkogel** | **mean** | **3.99** | **4.24** | **3.13** | **1.20** | **1.49** |
| | | | | | | |
| Taschach | 1 | 4.54 | 2.21 | 2.38 | 1.91 | 0.93 |
| Taschach | 2 | 3.78 | 2.39 | 2.27 | 1.67 | 1.05 |
| Taschach | 3 | 3.57 | 3.85 | 3.09 | 1.16 | 1.25 |
| Taschach | 4 | 4.12 | 3.97 | 2.17 | 1.90 | 1.83 |
| Taschach | 5 | 1.27 | 1.52 | 1.74 | 0.73 | 0.87 |
| Taschach | 6 | 1.82 | 3.84 | 2.11 | 0.86 | 1.82 |
| Taschach | 7 | 2.24 | 3.49 | 2.17 | 1.03 | 1.61 |
| Taschach | 8 | 2.08 | 1.95 | 1.77 | 1.18 | 1.10 |
| **Taschach** | **mean** | **2.93** | **2.90** | **2.21** | **1.30** | **1.31** |

Furthermore we determine the standard deviation. Therefore, we add to the manuscript:

in current line 221:
On average, seasonal variation is 1.30 (130%), with a standard deviation across all observations of 0.37 (37%).
and in current line 311:
At Vernagtferner, summer velocities are approximately 30% higher than the annual mean, with a standard deviation across all observations of 37%. The relatively high uncertainty results most likely from the fact that the absolute measured values over a month (July-Aug or Aug-Sep) have maximum values of about 50 cm. However, these values have a relatively high measurement uncertainty of +/-14cm (see chapter Measurement Uncertainty).

Line 200: The statement that it is "not possible to (semi-) automatically produce a reliable surface velocity map from aerial or satellite imagery for slow-flowing Vernagtferner …." is rather speculative, not taking into account capabilities of advanced airborne and spaceborne observation systems and analysis techniques. For example, several large satellite constellations with very high resolution SAR sensors are in space since several years. Some of these constellations provide repeat interferometric observations of excellent quality with repeat-pass intervals from one day onwards, as for example ICEYE InSAR products show.

Thank you for the helpful comment. We agree that the original wording "not possible" was too absolute and did not sufficiently take into account recent developments in high-resolution satellite and aerial remote sensing. We have therefore amended the passage and now refer to it as "challenging" to emphasize the difficulties, but not the fundamental impossibility.

In fact, there are now several satellite constellations with very high spatial resolution and short repeat intervals (e.g., ICEYE-InSAR products with daily repeats) that provide excellent data quality. Such systems open up new possibilities for detecting glacier movements. However, the specific problem of the Vernagt glacier remains: at maximum speeds of < 5 m a⁻¹, even with a repeat interval of one to several days, the displacement is only in the range of a few centimeters to millimeters, which is often within the signal noise in the currently available data. When using longer time periods, on the other hand, the melting process simply dominates the movement too much. We have added examples of other high-resolution satellite constellations to the text (see comment above).

Text:
It is highly challenging to (semi-)automatically produce a reliable surface velocity map from aerial or satellite imagery for the slow-flowing Vernagtferner with maximum velocities of less than 5 m\,yr$^{-1}$. Therefore, we selected a sub-region …

Line 223ff: Taking into account that the annual melt losses, amounting up to several meters, may cause significant changes of surface features, the estimates of feature position accuracy seem to be rather optimistic. For example, in line 53 it is stated that "surface features (e.g. crevasses) change considerably during this period", a possible source for increased uncertainties in feature tracking. Furthermore, oblique views sideways of the central flowline may introduce errors, in particular if the surface elevation at the time of the survey is not exactly known. Please provide information on the procedures in which way these issues are taken into account.

Thank you for pointing this out. We have amended the description accordingly. In the new version, we now also take into account possible lateral oblique measurements, which we estimate to add approximately 5 cm of further uncertainty. We now state the positional uncertainty at ± 35 cm. As noted by the reviewer, this estimate is optimistic, but we ensure that only suitable features are included in the analysis through manual case-by-case decisions and the targeted selection of stable features (e.g., unchanged column intersections). We have added this note to the text.

The uncertainty of localizing these features depends on the image resolution (<= 20 cm pixel size), the feature size (>= 40 cm), possible sideways oblique views of the features (depending on the respective angle, in the order of around 5 cm) and the co-registration accuracy of the images. Considering these parameters, the feature position uncertainty can be estimated at 35 cm. Manual case-by-case verification and explicit searching for crevasse intersection points ensure that only features that do not change too much are selected, thus guaranteeing this very high quality.

**Section 5.1, Ice dynamics**: Basic mechanisms related to slow-down of glacier flow are addressed, as well as possible causes for seasonal variations. However, (1.) quantitative estimates on the impact and magnitude of the different processes at the study glacier and

their (2) interactions during the observation period are missing. For contributing to the advancement of understanding of processes governing the slow-down of retreating glaciers, quantitative estimates would be essential. (3) Furthermore, hints on the significance of the study results in respect to the general glacier behaviour in this region would be of interest.

(1) Thank you for pointing this out. We have added a further analysis for the quantitative estimation of ice dynamics to the mass balance. This means that it is now possible to quantitatively estimate which process is the main driver at the Vernagtferner (ice dynamics or SMB). An assessment of the extent of the various processes is carried out.

**Appendix A: Ice dynamic**

[Figure]

**Figure A1.** a) areal surface massbalance summed for the years 2017 and 2018, b) surface Difference 218-2016, c) Difference between a and b, which is the vertical ice transport caused by ice dynamic. Background image: Orthoimage 2016 © Bavarian Academy of Sciences and Humanities (BAdW), 2016.

We add to the manuscript results:
To quantitatively estimate the magnitude of ice dynamics in relation to other processes, further analysis was carried out. This examines which parameter predominantly drives the VF, the SMB (surface mass balance) or the ice dynamics. The assumption is made that the surface difference between some years mainly reflects changes in the mass balance and the vertical ice transport resulting from ice dynamics. Provided that the surface difference and the areal mass balance are known, it is possible to estimate the influence of vertical ice transport in relation to the SMB. Both a surface difference and an areal SMB are only available for the period 2016-2018, which is a very short period from a glaciological point of

view. A more current surface model is not available and areal SMB has not been generated in earlier years. However, it can be assumed that this period basically reflects the current ice dynamics, as the glacier already has very low velocities during this period, similar to those observed in 2018--2023. The SMB, summed for the years 2017 and 2018, is shown in Fig. A1 a and is derived from observations for the direct, glaciological method of mass balance determination. The surface difference between the years 2018 and 2016 can be seen in Fig. A1 b. The difference between the two parameters gives the vertical ice transport caused by ice dynamics (Fig. A1c). The ice dynamics can be identified as follows: In the accumulation area (or former accumulation areas), negative values (red areas) indicate that ice masses are being transported downward from this location. The more negative the values, the greater the ice dynamics. In the ablation areas, on the other hand, positive values (green areas) indicate ice dynamics. Ice masses are transported to this area, so large positive values indicate high dynamics. Values around zero represent very low ice dynamics. An analysis of Figure A1 c shows that the Taschach and Brochkogel areas exhibit hardly any dynamics (values very close to 0). In contrast, dynamics are still present in the former accumulation areas of these two sub-areas. The two green areas in the Taschach area provide good control. These are rock islands that were not excluded from the areal mass balance. However, the rock islands have remained stable during the period and have not moved, which is why they appear in the vertical component of the ice dynamics.

A closer look at the Schwarzwand area reveals more ice dynamics. In the northern accumulation area of the Schwarzwand area (as well as on the Hochvernagt plateau), there are clearly positive values that cannot be explained by ice dynamics.  However, in the surface models, these areas are partially covered by snow and have hardly any features. It can therefore be assumed that the surface models in this area are highly uncertain and that these values are attributable to this. Overall, the higher dynamics of the Schwarzwand area may also be partly attributable to a possible slight tilt of a surface model in this direction. Furthermore, error values can be seen in the strongly negative northern areas of the Hochvernagt. These originate from the surface models, where an error has likely occurred in the neighboring aerial image sheet.

In order to enable a numerical estimation of the processes, which excludes outliers as far as possible, the median of the absolute mass balance (1.08 m/yr) and the absolute vertical ice dynamics (0.67 m/yr) were calculated.
Overall, further uncertainties must be taken into account, such as the compression and thus the change in altitude of possible firn areas due to a change in density.

We add to the manuscript discussion:
A quantitative estimation of the magnitude of the ice dynamics in relation to other processes for the period 2016-2018 has shown that the VF is currently mainly driven by SMB, with ice dynamics playing a minor role. Looking at the entire VF, the median absolute values for SMB (median = 1.08 m/yr) and vertical ice transport (median = 0.67 m/yr) suggest that the SMB process predominates.
Overall, there is hardly any ice dynamics left at the tongues, but there are significant changes caused by SMB. In the accumulation areas, on the other hand, there is still some ice dynamics, with a smaller SMB component. There are indications that there may be slightly more ice dynamics in the Schwarzwand area and that this area behaves

fundamentally differently from the rest of VF. This was already suggested by Reinwarth (1999) and can be confirmed here.

Due to a lack of data, it is not possible to estimate earlier dates. However, it can be assumed that in the early years, ice dynamics was the main driver of VF. The measured velocities were significantly higher, particularly around 1980, with an absolute SMB being significantly smaller than it is today.

(2) Thanks for the hint, we add an analysis of the temporal relationship between mass balance and velocity and compare it to other studies.

Stocker-Waldhuber et al 2019 have found evidence that ablation stakes can be well suited to reflecting the current status of a glacier. They show that the Kesselwandferner (also in the Ötztal valley) shows relatively direct response of ablation-area ice dynamics to changes in mass balance, regardless of the geometry of the glacier.

We have found evidence that the ablation stakes at VF also react quickly to a change in the mass balance. Only with a slight decay, the strongly negative mass balance in 1983 appears to lead to a decrease in velocity in the ablation area. This could be attributed to the fact that ice thicknesses at the tongue (in the ablation area) are usually already very low. Even small changes in ice thickness can have a noticeable effect on velocity here, as the driving stress is highly sensitive to ice thickness. In a year with a strongly negative mass balance, the strong melt in the ablation area directly decreases the ice thickness, which represents the current status of the glacier. On VF, there is insufficient data to substantiate this adequately. Further studies (including in other areas) are necessary to investigate this effect in more detail.

(3) Unfortunately, apart from Vernagtferner, Hintereisferner and Kesselwandferner (we added information on that, see above) there are no other long-term measurements available for this region. Since they are two significant glaciers in the region with similar changes in ice dynamics, it can be assumed that the other glaciers in the region behave similarly, but we do not have any data on this. Instead, a comparison was made to a wider regional trend.

text:
Neighboring glaciers, such as Hintereisferner show a similar velocity peak around 1980, followed by a decrease (Stocker-Waldhuber et al., 2019). The velocity peak aligns with a wider regional trend, as a significant proportion of glaciers in the European Alps experienced advances during this temporarily relatively cool phase (Patzelt, 1985; Wood, 1988).

---

## Author Comment (AC2)

**Author´s response to referee comment # 2**

**First of all, we would like to thank the reviewer for such a detailed and constructive review. It has shown us which aspects need to be explained in more detail and has certainly helped us to improve the manuscript. Bevor responding to every comment, we would like to mention, that we adapted the manuscript in the following points to ensure a better readability:**

- **We will follow the reviewers recommendation to restructure the current chapter 2 "Methods and Data".**
- **We fully agree that, based on our tests, automated derivation of surface velocities cannot be ruled out (the manuscript will be adapted accordingly). We would rather point out the high ablation, which overlays the ice dynamic feature change, and thus explain why we choose manual feature tracking.**
- **The reviewer suggested to skip parts of the manuscript by focusing either on historical data or on the interpolated present-day velocity map. As mentioned, we don't want to rule out automated derivation of surface velocities. The focus of the manuscript is more on the historical data sets, whereby we believe that an examination of current ice dynamics is relevant in order to provide a velocity map ready for other studies, e.g. ice flow modelling**

In the following, we quote the reviewer´s comments followed by our replies, which are marked in orange. Since the reviewer posted many comments (i.e. general, major, specific with several subpoints), we tried to organize that with a kind of chapter structure to make cross-referencing easier. His/her general comments receive chapter index 1, while major comments chapter index 2, with every subpoint indexed accordingly. Sub-items within a comment are also numbered with numbers in brackets.

**General comments**
The manuscript by Dobler et al. analyzes the ice surface velocity of Vernagtferner, as an example of a well-monitored slow-flowing mountain glacier.
Leveraging 60 years of annual observations of stake positions, the manuscript draws a link between the glacier mass balance history and the patterns of ice speed-up and slowdown. After concluding that existing remote sensing products fail to resolve the slow flow field of Vernagtferner, the manuscript presents a map of present-day surface velocities over 2018-2023, compiled by interpolation of multi-source point measurements such as stakes and manually tracked features.
The topic of remote and in situ ice velocity measurements on slow-flowing glaciers is certainly of current interest, and a report on the long-term dynamics investigations at a data-rich site is definitely relevant and holds significant potential to advance knowledge of the ice dynamics of mountain glaciers.
However, the main direction of the manuscript is not fully clear in its current form - the two covered topics (re-analysis of long-term stake data and compilation of present-day velocity map) are only weakly linked. As such, the main message and achievements of the manuscript are somewhat hard to understand. Moreover, the manuscript stops short of advancing the current knowledge on the topic of ice dynamics. The glaciological conclusions

from the analysis of such a rich historical dataset are somewhat qualitative and generic, and in some cases are not adequately supported by the collected evidence. Much of the data needed for interpretation of the ice dynamics (such as historical local mass balance, or changes of ice thickness and surface slope) are not adequately presented in the manuscript. Furthermore, the proposed method to compile a velocity map (in part) from manually tracked point measurements is affected by some significant flaws, raising questions about its suitability and advantages compared to state-of-the-art automated methods. Finally, the uncertainty analysis relies on several arbitrary estimations and assumptions, rather than existing, well-established methods to quantify uncertainty in remotely sensed glacier dynamics.

1.1 Thank you for pointing this out. We will explain these points in detail below.
- Redefine research question see response to general comment 1.2.
- Strengthening glaciological interpretation and additional supporting evidence, see response to major comment 2.5.
- Add missing datasets, see response to major comment 2.1(4).
- Address methodological weakness in velocity map construction see response to major comments 2.3 and 2.4.
- regarding uncertainty analysis see response to major comment 2.6.

In light of this, I would suggest to repeat some key parts of the analysis, after reviewing the literature for the most appropriate methods to be applied to the high-quality datasets of Vernagtferner. I would suggest resubmitting the manuscript to reflect these major changes. In particular, I would suggest adjusting the scope to focus more on just one of the two topics - either (i) the compilation of the present-day velocity map, or (ii) the analysis of the historical dataset and its glaciological interpretation. The two topics are quite loosely related and the manuscript would really benefit from a clearer message, answering one or more well-defined research questions.

1.2 We define our research question more clearly. The aim of the manuscript is not to rule out the automated creation of a velocity map (we will reword the relevant sections of the manuscript accordingly), but rather to highlight the problems/challenges associated with high ablation for retrieving glacier velocities on slow-moving glaciers with remote sensing techniques. However, your raised points in this response convinced us to explain in more detail. We try to better explain what we mean by the problem of "high ablation" in the context of feature tracking (detailed information on this can be found in the response to major comments 2.3). Overall, the manuscript focuses more on (ii) the analysis of the historical data set, and we have also expanded the analyses to reflect this. However, we believe that detailed information on current ice dynamics is also highly relevant when analyzing historical data, especially since high-resolution data is available for the current period. Since the dynamics are now only minor, the maximum remaining velocities are particularly relevant. We think, generating a present-day velocity map is a logical consequence based on the data availability in order to provide a velocity map ready for other studies, e.g. ice flow modelling (either used as validation as target for an inversion).

For topic (i), I would specifically suggest application of digital image correlation on the UAV, aerial and/or satellite imagery, with an appropriate pipeline for pre-filtering, post-processing, and aggregation. Multiple studies (e.g., [1], [2]) have shown that data processing specifically optimized to a study site can resolve the ice dynamics of individual glaciers much better than global products. Given the high-quality available data (such as four years of end-of-season, whole-glacier airborne photogrammetry), I expect this processing to resolve very well the

slow movement of Vernagtferner and automatically produce good-quality velocity maps, possibly even at annual intervals and on the whole glacier body, contrary to the current manuscript's conclusion that "It is obviously not possible to (semi-)automatically produce a reliable surface velocity map from aerial or satellite imagery for the slow-flowing Vernagtferner". The stake measurements (and possibly the manually tracked feature) would be a valuable reference for validation and uncertainty estimation. This kind of manuscript might be most suitable for a data-focused journal.

1.3 As mentioned, the focus should not be on excluding possible (semi-)automatic detection methods for the Vernagtferner. This statement will be removed from the manuscript, as we don't test all possible methods. Instead, we would like to point out the special features of (automated) detection with high ablation. More detailed information on this can be found in the response to major comments 2.3.

For topic (ii), I would specifically suggest a more thorough re-analysis of the very interesting historical stake data, possibly including: (1) a more realistic model approach than the shallow ice approximation, for example an IGM inversion, given the high-quality data available over the entire glacier; (2) a better-processed ice thickness map, without the obvious major artifacts visible in Fig. 7; and (3) a more detailed analysis of the interplay of surface slope, glacier thickness, and stake velocity anomaly. The recent measurements shown in the manuscript could still be mentioned to compare spatial patterns over time and to investigate seasonality. This kind of manuscript would be most suitable for a glaciological journal.

1.4 Thank you very much for your valuable suggestions for further analysis. Regarding:

(1): You are right, a more sophisticated model would be appropriate for modelling the response of VF with the observed data for validation/tuning/inversion. However, our intention was to estimate whether the observed dynamic trends in the historical velocity data are connected to mass balance trends (i.e. ice thickness). We think for this rough estimation a simple model (i.e. SIA) is a fair approach, particularly, as our paper is not focusing on modeling. Using an IGM inversion would be a good approach but shifts the focus more to modelling rather than on the observations. The data on ice thickness and surface slope for the individual years are heavily interpolated (more on this under response to major comments 2.1 (1)), making realistic modeling difficult based on the initial data. However, the focus of the manuscript is on showing the velocities over time. Therefore, we only want to use SIA modeling to generate an uninterrupted timeline of the geometry in order to show that the values are realistic in principle. This is therefore not an ice dynamics model in the true sense, but rather a first approximation. We will explain this explicitly in the manuscript.

(2): The artefacts visible in Fig. 7 have been corrected (for result see the response to minor comment of Fig.7) . The calculation of ice thicknesses is explained in more detail under response to major comments 2.1 (1).

(3): Discussion of the analysis of the anomalies of various parameters such as ice thickness and surface slope can be found under response to major comments 2.5.

In both cases, it is important to perform a quantitative, data-driven uncertainty analysis, based on leave-one-out validation and rigorous error propagation, as described in the references provided below.

Detailed information under response to major comments 2.6 and specific comments.

**Specific major comments:**
- Presentation of the Data and Methods needs major restructuring in order to support the analysis. In the current form, it is really hard to understand where some important data come from or how they were processed. In particular, it is important to provide full information on: (1) the data sources used for glacier thickness, as well as all processing applied to (i) extrapolate it to the full glacier area, (ii) compute evolution over different time periods, and (iii) extract it at the stakes location; (2) all the stakes (at the very least, all the stakes whose data are plotted in the figures), with their identification numbers, observation period, and maximum/minimum altitude; (3) the actual calculation of modeled velocities according to the shallow ice equation, detailing the estimation and assumptions made for each variable; and (4) the processing of UAV and aerial data, including georeferencing, the used software pipelines, and the resulting spatial resolution and accuracy. Some relevant pieces of information are already mentioned in the manuscript, but in a rather scattered form across the sections, and should be rearranged.

2.1 Thank you for pointing this out. We reconstructed the Methods and Data section.
(1): A flow chart as well as the following explanation for calculating ice thickness and surface slope for each year was added:

[Figure]

To provide an ice thickness for each averaged period, the surface change between the known periods (DSMs) was scaled using the mass balance data at each date and the known

glacier bed topography was subtracted. The mass balance is described by a single value per year, as area-based mass balance is not available for all periods. The error of non-linear ice transport over time is neglected, as the velocities are relatively low. The calculated ice thickness and surface slope is cut to the least known glacier extent, an overview of the existing extents is in the Appendix.
The extraction on the stake position is carried out through the nearest neighbour.

(2): An overview of the measurement points that appear in the figures will be added to the appendix.

3): A chapter on the shallow-ice approximation modeling, including formulas used and estimated parameters, will be added:

We add a new paragraph to the data and methods section

**shallow-ice approximation**

Using a simplified shallow-ice approximation, the surface velocity can be derived from the surface slope ($\alpha$) and ice thickness (H) (e.g. Hutter 1983, Grever and Blatter 2009). This is also referred to as a 'zero-order' model, whereby only vertical shear stress gradients are taken into account. The total velocity ($u_{total}$) consists of the deformation velocity ($u_{deform}$) and the basal sliding velocity ($u_{basal}$). We only want to use SIA modeling to generate an uninterrupted timeline of the geometry in order to show that the values are realistic in principle. This is therefore not an ice dynamics model in the true sense, but rather a first approximation. Since the assumption of uniform basal sliding over a period of more than 30 years is certainly not accurate, and the velocities in recent times are relatively low anyway, making it difficult to distinguish between basal sliding and deformation velocity, a basal sliding velocity of 0 was assumed for simplicity and the modeled velocities were calculated purely on the basis of ice dynamics. Relatively soft ice was used for this purpose, therefore we use for the rheology parameter A a value corresponding to 0 degrees, according to Paterson 1994. We just want to show the link between geometry and ice dynamics.

$(u_{deform}) = (2*A / n+2)) * \tau^n * H)$

, where $\tau$ is the shear stress, described as:

$\tau = -\rho * g * H * \sin(\alpha)$

The selected physical flow parameters are summarized in the following table:

| parameter | value | unit |
|-----------|-------|------|
| A | 2.2e-16 | Pa^-3 a^-1 |
| n | 3 | |
| $\rho$ | 900 | kg/m³ |
| g | 9.81 | m/s² |

(4): An overview of all available data is added as an appendix:

| dataset | date of acquisition | sensor/instruments | resolution [m] | processing | coverd area | source / access | dataprovider |
|---|---|---|---|---|---|---|---|
| stake measurement | yearly, end of september [1966-2023] | multiple forward intersections, tachymetric polar coordinate or GNSS | / | remove gross errors | entire glacier, mainly ablation area | https://doi.org/10.1594/PANGAEA.854595 | |
| | 2022-08-04, 2022-07-10, 2020-09-21, 2023-08-08 | multi-frequency GNSS receiver | / | post-processing with fixed ambiguities | | https://doi.org/10.1594/PANGAEA.854595 | |
| glacier bed topographie | 2006 & 2007 | Radar transmitter with antenna | 20 | see Mayer et al 2013 | entire glacier | zenodo DOI | Mayer et al 2013 |
| Orthofotos | 2022-07-10 and 2022-09-21 | UAV DJI with optical RGB camera | 0,04 | Georeferenzierung with ground control points, orthophotogenerierung in Agisoft Metashape | Parts of Taschach and Brochkogel | zenodo DOI | |
| | 2018-09-16 | Optical airborne photogrammetry, UltraCam Eagle Mark 2 | 0,2 | Aerotriangulation: Inpho(Match-AT) by Trimble, orthofoto generation: SURE (nFrames) | entire glacier | publication do to legal rights not possible | Geissler 2021 |
| | 2020-09-08 | Optical airborne photogrammetry | 0,2 | see tiris Tirol | entire glacier | tiris Tirol online | |
| | 2021-09-25 | Optical airborne photogrammetry | 0,2 | information not available | entire glacier | publication do to legal rights not possible | |
| | 2022-08-23 | Optical airborne photogrammetry | 0,2 | information not available | entire glacier | publication do to legal rights not possible | |
| | 1969 | aerial photogrammetry | 20 | information not available | entire glacier | zenodo DOI | |
| | 1982-09-14 | aerial photogrammetry | 20 | information not available | entire glacier | zenodo DOI | |
| | 1990-08-24 | aerial photogrammetry | 20 | information not available | entire glacier | zenodo DOI | |
| DSM | 1999-09-09 | aerial photogrammetry | 20 | information not available | entire glacier | zenodo DOI | Weber 2013 |
| | 2009 | Laserscaning | 1 | information not available | entire glacier | zenodo DOI | |
| | 2016-09-29 | aerial photogrammetry, Canon EOS 5D Mark II | 1 | Photogrammetic analysis: PIX 4D 4.2.27 | entire glacier | zenodo DOI | |
| | 2018 | aerial photogrammetry, UltraCam Eagle Mark 2 | 2 | Aerotriangulation: Inpho(Match-AT) by Trimble, DSM generation: SURE (nFrames) | entire glacier | publication do to legal rights not possible | Geissler 2021 |
| glacier outline | 1969,1979,1990,1999,2009,2016,2018 | | / | information not available | entire glacier | zenodo DOI | |
| mass balance | yearly 1966-2024 | glaziological mass balance | / | | entire glacier | WGMS | |
| global dataset ice dynamic | | ITS_LIVE | / | see Gardner et al 2022 | entire glacier | | Gardner et al 2022 |
| | | FAU's glacierportal | / | see Friedl et al 2021 | entire glacier | | Friedl et al 2021 |
| | | Millan's dataset | / | see Millan et al 2022 | entire glacier | | Millan et al 2022 |
| satelite SAR imagery | see Appendix C | Terra-SAR-X | 2 | GAMMA Remote sensing Software | entire glacier | | |

Literature/Links to the table:

- Weber 2013: Dokumentation der Veränderungen des Vernagtferners anhand von Fotografien
- Mayer et al 2013: Vermessung und Eisdynamik
- tiris Tirol: https://lba.tirol.gv.at/public/karte.xhtml
- Geissler et al 2021: Analyzing glacier retreat and mass balances using aerial and UAV photogrammetry in the Ötztal Alps, Austria

Unfortunately, much of the information, such as the software pipelines used or the georeferencing of the source data, is unknown. Where possible, references to the respective source are provided, but particularly regarding the historical surface models there is no known information. As long as there are no legal reasons to the contrary, the data sets are going to be made publicly available via the Zenodo platform. This information can also be found in the table.

- The discussion lacks almost any comparison of the findings of the present study with those of other investigations, in particular those about (1) long-term trends of ice velocities at reference glaciers in the European Alps, which are monitored at several key sites across the Alpine countries (a single publication on Hintereisferner is currently cited); (2) the derivation of velocity maps of slow-moving glaciers from remote sensing (e.g., [2], [5]); and (3) previous examinations of the dynamics of Vernagtferner (ll. 91-92). It is crucial to situate the findings of the current study in the context of published results, in order to highlight advancements and challenges.
2.2 (1): That´s true. We are expanding the discussion:

In the long-term observation of the d'Argentière glacier, Mont Blanc area, Vincent et al 2009 show an ongoing negative net mass balance since 1982, with a direct reflection (delayed by a maximum of 3 years) in the velocity. They describe that the change in velocities in the upper part of the glacier is smaller than in the lower part (ablation area). The velocity trend thus fits the VF timeline quite well. A larger change in velocities in the ablation area suggests that the current status of the glacier can be derived relatively well, especially at the ablation stakes, as we suggest for the VF.

see also expansion due to discussion of possible "direct response", and compare to studies in response to major comment 2.5.

2): Thank you for pointing that out. We are expanding our discussion regarding the derivation of velocity maps for slowly flowing glaciers, as follow:

Even for relatively slow-flowing glaciers, such as the Griesglacier, surface velocities can be derived using software such as IDMatch, a tool for automated velocity derivation (Gindraux 2019). Compared to the VF, the average velocities on Griesglacier with more than 1 m in just 1 month are significantly higher than the average velocities at the VF of about 1 m per year. Furthermore, a short test of the IDMatch software using the UAV VF datasets from July 2022 and September 2022 does not provide any results, as no correlations can be found. Regardless of the software and the possible detection of identical features, for VF there remains a strong overlap due to ablation (see Appendix A in response to major comment 2.3), thus distorting the results. We do not want to rule out the possibility of automated derivation, but it would be essential to additionally correct for the false values caused by the ablation-induced change in features (a change not caused by ice dynamics). For this reason, we have chosen manual feature tracking. Examples of the differentiation between ablation-induced feature changes and ice dynamics-induced feature changes in manual feature tracking are presented in Appendix A (in response to major comment 2.3).

(3): The two current studies show the changes in velocity up to the respective point in time, with basic analysis. Since this information is already contained in the manuscript, these studies will not be discussed in detail. However, since the data preparation up to the respective point in time and a rather rudimentary display were carried out here, we would like to mention their valuable work at this point. To avoid misunderstandings, we rewrite the sentence to: The derived velocities have been displayed in previous studies up to the respective point in time.

- The manuscript claims that "It is obviously not possible to (semi-)automatically produce a reliable surface velocity map from aerial or satellite imagery for the slow-flowing Vernagtferner", which is the main motivation to propose a manual tracking method (with the resulting downsides for spatial coverage, reproducibility, and labor effectiveness). However, this conclusion is based on observation of some global datasets of glacier velocity, as well as some very poorly detailed testing of automated methods on the available datasets, which are quickly dismissed as an option (ll. 113-119 and 189-199). However, several studies have thoroughly validated the derivation of glacier velocity fields on aerial and UAV imagery with automated methods, such as frequency-domain cross correlation ([3], [4]) and feature tracking ([5], [6]). These methods have been shown to resolve well the ice motion on optical imagery, typically with subpixel accuracy; in particular, the result of [5] (p. 58) clearly shows UAV-based feature tracking fully resolving ice displacements of the same magnitude and time interval as those of Vernagtferner. Thus, a claim of unsuitability of those well-established methods needs to be supported by much better evidence than a quick dismissal, especially given the high-quality available datasets at Vernagtferner (UAV and aerial imagery). The interest of manually pinpointed displacements of individual features is hard to justify without first testing and quantitatively reporting on these state-of-the-art methods.

2.3 You are absolutely right; we cannot rule out automatic methods at this point. We will amend the manuscript accordingly. The aim of the manuscript is not to test all algorithms and rule out the possibility of automation. Instead, we would like to highlight the problems of overlapping due to ablation from a more glaciological point of view and explain why we chose manual detection. We would like to show that detection is not "rudimentary" and cannot simply be carried out using global data sets or standard remote sensing methods. Thank you very much for the reference [5] to a study on the detection of slow velocities. The study shows an average surface dynamics of more than 1 m over a period of about 1 month, with maximum values of up to 3 m. On Vernagtferner, the average velocity is only 1 m per year, with maximum values of 4 m per year. Over a period of one month, Vernagtferner will therefore only reach values of approx. 10 cm to maximum values of approx. 40 cm. This means that the dynamics of the Vernagtferner are once again slightly lower than those of the Griesglacier shown in the study. Furthermore, we only have high-resolution UAV data for the dates 2022-07-10 and 2022-09-21, i.e., for a temporal baseline of 2.5 months (the other datasets have significantly longer baselines). During this period, there was a significant change in surface structures, mainly due to high ablation. Nevertheless, we tested the IDMatch software for the UAV data. Unfortunately, no correlations could be found. This information will be added to the manuscript, and we will also mention further software options. (See also response to major comment 2.2 (2).)

The UAV measurements in 2022 relate to a year of extreme melting. This melting also causes significant changes to surface features. For this reason, we write in the manuscript that "high ablation means that the horizontal melt is even greater than the actual movement." We did not go into further detail on this. The feedback has shown us that we need to clarify what exactly we mean by changes in features due to ablation. To this end, we would like to include the following examples in the appendix A:

The figures show the same situation at different points in time:

[Figure]

Fig. 2: Melt-induced changes to crevasses. The example is taken from the upper part of the Taschach area at an altitude of approx. 3150 m. Maximum velocities occur in this area at the VF.

Figure 2 clearly shows a change in surface features, in particular a widening of the crevasses. We examined two of the crevasses in more detail using two stakes at the upper and lower edges of each crevasse. The average movement of the eight stakes (shown in Fig. 2) over the two months is approximately 0.73 m (change from the red to the blue points), with a standard deviation across the eight stakes of 0.12 m . The stakes rule out the possibility that a significant actual break-up of the crevasses has taken place. The stakes

show an even shift of the upper and lower edges of the crevasses. Thus, the change in the surface is probably largely due to melting. Even if identical features (in this case, crevasse edges) could be identified during the period, the movement would be significantly overlaid by ablation due to the crevasse edges (and their varying melt rates), resulting in erroneous dynamics.

Even with larger temporal baselines, manual feature tracking can be used to exclude features that are likely to have changed due to ablation, as is often the case in 2022. Crevasse trace intersections are particularly well suited for this purpose, as can be seen in Fig. 3. Identifying identical features that have not been subject to high ablation is challenging even for the human eye, especially over longer baselines, such as the example in Figure 4 over a period of two years.

[Figure]

Fig. 3: Melt-induced changes to crevasses. Example from the upper part of the Brochkogel area at an altitude of about 3200m.

[Figure]

Fig. 4: Crevasse pattern in a period without significant melt. Example from the highest part of the Brochkogel area at an altitude of about 3250 m.

- The 2018-2023 velocity map is aggregated from data collected over 5 years, on the assumption that ice dynamics would "not change much" over that period. However, Fig. 4 and Fig. 5 show ice velocity changes by more than 50 % taking place over 5 year periods at several stakes. Such a change is larger than (for example) the 30 % correction factor applied

to convert summer to annual stake velocities in the data used within the velocity map. As such, the validity of the aggregation of such heterogeneous data is questionable and should at least be discussed in the uncertainty budget. Moreover, the assumption of zero velocity at the glacier edges is questionable - a contribution from transverse stress coupling could potentially be significant on such a slow-flowing glacier. See for example [7]. A rigorous analysis and discussion of these uncertainties is required to support the presented results, especially when the manuscript claims that other existing methods and results are not suitable for the study site.

2.4 Thank you for pointing that out. The assumption of an "unchanged" velocity for the period 2018-2023 needs to be explained in more detail and substantiated with data. We used the stake series (measured over at least three epochs in the period 2018-2023) to calculate whether there was a significant average slowdown in velocity from one epoch to the next, taking into account the measurement uncertainty. As explained in the manuscript, the measurement uncertainty of the stakes is +/-10 cm or, taking into account the law of error propagation, +/- 14 cm for the calculated velocity.

[Figure]

Fig. 5: Average variation in velocity between the periods 2018–2023, for stakes with at least three measured values during this period.

There are no measured values for the period 2018-2019. The variation from 2019 to 2022 is within the measurement uncertainty. Only the period 2022-2023 shows a decrease of approximately 10% (excluding the measurement uncertainty), but there are only three relatively widely scattered measurements for this period. Therefore, it cannot be assumed that there is a significant decrease. However, this uncertainty should be taken into account in the uncertainty budget by including the maximum variation of 10% in the measurement period 2019-2023. We will add the maximum variation in the uncertainty chapter of the manuscript.

We agree that transverse stress couplings are not taken into account when assuming a velocity of 0 m/yr at the glacier margin. Nevertheless, the dynamics at the margin of the glacier are significantly slower than in the center. Since the VF itself flows relatively slowly even in the center, the velocities at the margin will be close to 0 m/yr. The reference to transverse stress couplings will be included in the manuscript.

- The manuscript mentions a "strong sensitivity of velocity to mass balance", claiming that "even the small effect of slightly positive mass balance years on the glacier geometry, results

in very pronounced changes of ice flow". While this "effect [...] on the glacier geometry" is not further described, this conclusion suggests that changes of glacier geometry due to positive glacier-wide mass balance would be reflected as faster ice flow in the ablation area already during the same mass balance year, with little to no lag. Such a conclusion is somewhat at odds with the notion of glacier response time and would need to be backed up by some evidence (such as an actual analysis of the mentioned "effect on glacier geometry"). As such, a more thorough analysis of the interplay between glacier mass balance, geometry changes (especially thickness and slope), stake location within the glacier, and stake velocity, is needed.

2.5 We agree that a more detailed analysis of anomalies in, for example, glacier geometry and stake velocities could reveal the sensitivity to the mass balance more accurately. The parameters surface slope and ice thickness for each year are derived from an interpolation, whereby a surface model is only available approximately every 10 years. The scaling of the change in surface area between years is performed using the mass balance (as described in more detail under response to major comment 2.1(1)). However, a continuous stake series extends over a maximum of two of these known surface models before the stake melted out. The remaining surface models and ice thicknesses are derived from interpolation. Accordingly, an investigation of the anomalies is strongly influenced by the interpolation, and this in turn is influenced by the mass balance, among other factors. In order to avoid generating a false dependence of the mass balance on the glacier geometry (slope and ice thickness), we focused on the mass balance parameter and did not carry out any investigations into the glacier geometry.

Furthermore, we fully agree that the effect we describe as "strong sensitivity of velocity to mass balance" suggests the conclusion that changes in glacier geometry due to mass balance are reflected in the ablation area with little or no delay. As already mentioned, we are unable to perform more detailed analyses due to the rough interpolation of the geometry data. The effect can be seen particularly clearly in Figure 4 of the manuscript for the year 1983. Already in the year following a strongly negative mass balance, the velocity appears to decrease significantly. Unfortunately, very little stake information is available for the years 1983-1985, which are particularly relevant here. However, this information suggests that the stakes in the ablation area react relatively directly to changes in mass balance. We will now describe this effect in more detail and discuss it with the help of further literature:

Stocker-Waldhuber et al 2019 have found evidence that ablation stakes can be well suited to reflecting the current status of a glacier. They show that the Kesselwandferner (also in the Ötztal valley) shows relatively direct response of ablation-area ice dynamics to changes in mass balance, regardless of the geometry of the glacier.

We have found evidence that the ablation stakes at VF also react quickly to a change in the mass balance. Only with a slight decay,the strongly negative mass balance in 1983 led to a decrease in velocity in the ablation area. This could be attributed to the fact that ice thicknesses at the tongue (in the ablation area) are usually already very low. Even small changes in ice thickness can have a noticeable effect on velocity here, as the driving stress is highly sensitive to ice thickness. In a year with a strongly negative mass balance, the strong melt in the ablation area directly decreases ice thickness, which represents the current status of the glacier. On VF, there is insufficient data to substantiate this adequately.

Further studies (including in other areas) are necessary to investigate this effect in more detail.

- Most uncertainty estimations (Sect. 4.4) appear to be qualitative, arbitrary or statistically inaccurate. A more rigorous uncertainty analysis is needed, since an extensive literature exists on relevant methods, specifically concerning glacier dynamics (e.g., [8], [9], [10]). The availability of a large number of data points suggests application of a leave-one-out method for robust uncertainty estimation. Finally, the formulas used to calculate and transform uncertainties should be shown.

2.6 Thank you for pointing that out. In this section, we also address your minor comments on line 229 and 235.

As suggested in reference [14] (minor comment on line 235), the interpolation uncertainty can be calculated using a bootstrap technique. We tried this technique on our interpolated velocity field consisting of 177 measurement points. To do this, we selected 10 random datasets from 20, 50, and 80% of the points and interpolated each of them. At the skipped points, the misfit to the measured velocity was plotted (separately for velocity in the X and Y directions). The relative misfit ( the absolute misfit ($\Delta v$) in relation to the measured velocity ($v_{x\,meas}$)) in percentage is shown as gray points in Figure 90. The median is shown in red, as well as the $2\sigma$-confidence interval, grouped for distance intervals. Sufficient measurements are only available for the range up to approx. 50 m distance to the closest observation. A significant trend, as could be described by a linear regression, for example, cannot be found.

This is most likely due to the inhomogeneous distribution of the points and the fact that there is a high point density with a high variation in magnitude in the crevassed areas, whereas there is a low variation with a low point density on the glacier tongue. Due to the structure of these data, no statement can be made about a possible dependence of the interpolation uncertainty on the distance to the nearest observation point.

[Figure]

[Figure]

Figure 90: Misfit between modeled and measured velocity ($\Delta$ v) in relation to the measured velocity (v $_{meas}$) versus distance to the nearest observation as grey points. Red points indicate the median for a distance interval with its 2σ-confidence interval.

Furthermore, we performed a leave-one-out method. The results (Figs. 91, 92, and 93) and a description, will be added to the manuscript as Appendix B as follows:

A further uncertainty analysis can be performed using a leave-one-out method. For this purpose, an interpolation was performed for each point without using this point and the misfit of the interpolated velocity to the measured velocity was determined. The generated misfits are shown as relative and absolute errors with respect to the distance to the closest observation in Figures 91 and 92. The figures show that there is no significant correlation between the misfit and the distance to the nearest observation. This is most likely due to the inhomogeneous distribution of points and the fact that there is a high point density with a high variation in magnitude in the crevassed areas, whereas there is a low variation with a low point density on the glacier tongue. This can be seen in Figure 93. The individual misfits for each point are shown, with the mean average being formed in case of an overlay. The percentage uncertainty does not appear to be significant in specific areas. Due to the structure of the data, no statement can be made about a possible dependence of the interpolation uncertainty on the distance to the nearest observation point. Instead, as can be seen in Figures 91 and 92, there is a normal distribution around the value 0, so a 1σ-confidence interval can be derived as the mean uncertainty of the interpolation  (in x- and y-direction). The relative error is estimated at 50.4% in x-direction and 28.3% in y-direction, while the absolute error is 0.6 m/yr in each direction.

The interpolation uncertainty (u $_{interpolation}$ ) for the velocity magnitude can be calculated from the 1σ-confidence interval in x- and y-direction as follow:

$$u_{interpolation} = \sqrt{u_{interpolation\,x}^2 + u_{interpolation\,y}^2}$$

resulting in an interpolation uncertainty of 57%, respectively 0.8 m/yr.

[Figure]

Figure 91: Misfit between modeled and measured velocity (Δ v) in relation to the measured velocity (v $_{meas}$) versus distance to the nearest observation according to a leave-one-out-method, referred to as u $_{interpolation\_x}$ and u $_{interpolation\_y}$.

[Figure]

Figure 92: Misfit between modeled and measured velocity (Δ v) in relation to the measured velocity (v $_{meas}$) versus distance to the nearest observation according to a leave-one-out-method, referred to as u $_{interpolation\_x}$ and u $_{interpolation\_y}$.

[Figure]

Fig. 93: Interpolation uncertainty (u $_{interpolation\_x}$, u $_{interpolation\_y}$) as misfit of a leave-one-out-method, with the misfit between modeled and measured velocity (Δ v) in relation to the measured velocity (v $_{meas}$) for measuring point, averaged in case of overlaps. a) for velocity in x-direction, b) for velocity in y-direction .

Furthermore, a position uncertainty (referred to as u $_{measured}$) of the measured points can be calculated for the velocity, by following the covariance propagation law. Where t is the temporal resolution in years and $u_{startdate}$ the position uncertainty of the startdata, as well as

$u_{enddate}$ for the position uncertainty of the enddata, from whom the velocity of each point is calculated. The start and end date, the temporal resolution as well as the resulting velocity uncertainty for each set of data is reported in Tab.3.

$$u_{measured} = \sqrt{(u_{startdate}^2 + u_{enddate}^2)/t}$$

Regarding the utilized datasets, different measurement uncertainties arise. An approximate total measurement uncertainty $u_{measured\ total}$ across all observations can be estimated:

$$u_{measured\ total} = \sqrt{\sum_{i=1-5} (n_i * u_{measured\ i}^2) / n_g}$$

with i indicates a row in Tab. 3 and $n_i$ being the number of points of the respective dataset, $n_g$ the total amount of points. This leads to an approximate total measurement uncertainty of 0.27cm.

For clarification in the main manuscript we:

-delete line 228-230, as the mean uncertainty is not the mean of individual uncertainties.

- will change the column name of column 5 to: position uncertainty $u_{measured}$

- add in line 226: The resulting velocity uncertainties represent one standard deviation (1σ) and are listed in Tab. 3, a detailed description to the calculation is in Appendix B. It is also demonstrated that an approximate total measurement uncertainty of 0.27 m/yr can be estimated.

- add in line 233: To examine the uncertainty of the interpolation more closely, a leave-on-out method was performed. This shows that there is no significant dependence of the misfit (between modeled velocity and measured velocity) and the distance to the next observation. However, a relative interpolation uncertainty of 57% and an absolute interpolation uncertainty of 0.8 m/yr can be estimated from the standard deviation. Detailed calculations and explanations can be found in Appendix B.

**Specific minor comments:**
- The Introduction needs to provide more focused context on the specific topic of the manuscript, citing more literature on the monitoring of slow-flowing mountain glaciers, on long time series of ice velocities, and on the creation of glacier velocity maps for single glaciers; at present, it is somewhat meandering over broad topics of ice dynamics (basics of glacier flow, glacier hydrology, global ice velocity products).

That´s true. We provide further specific context in the introduction:
to current line 27:
Therefore, long-term measurements are essential for detailed analyses. A long time series of measurement make it possible to investigate the flow processes of a glacier and their changes due to the glacier response to climate change, taking into account the glacial processes. Vincent et al. 2016 were able to show that for the Argentière glacier in the Mont Blanc area, no correlation can be found between changes in surface velocity and subglacial water runoff. A change in mass balance, on the other hand, has a direct influence on the ice dynamic behavior of this glacier, as Vincent et al. 2009 find a direct or delayed reaction of maximal three years. However, phenomena are also found that deviate from a correlation between surface geometry and velocity change. For example at glacier de Saint Sorlin, France, velocities around the year 2000 are still greater than in 1960, despite a negative cumulative net mass balance since 1957 (Vincent 2000). This shows the complexity of the processes.
Spatial high-resolution velocity information as well as long term monitoring allow a detailed and precise analysis of glacier behavior.

We add in line 50: For individual glaciers, better results can be achieved with specific optimized data processing than with global data sets (Mattea 2025). Even relatively low ice velocities can be detected, as Gindraux 2019 shows for Griesglacier. However, slow-flowing glaciers need a specific temporal baselines between image pairs to capture recognizable flow. This implies a potential loss of coherence, as surface features (e.g. crevasses) can change considerably during this period (van Wyk de Vries and Wickert, 2021).
- l. 29, Nye (1959) was most definitely not the first rigorous investigator of ice velocities; see e.g. [11]
That´s true. We change it to "A method established by Nye(1959)...".
- l. 48, what are "sensors such as Sentinel-2" compared to "other optical sensors"? Define the groups or reword for clarity.
We reword for clarity: "Millan et al (2019) showed that Sentinel-2 data can produce more precise results than Landsat-8."
- l. 69, "continuous monitoring" is unclear given the present-day availability of automated, sub-hourly monitoring sensors. Consider using "systematic" or similar
Thanks for the hint, we will use "systematic" instead of "continuous".
- l. 72, what makes the site "unique"? Explain concrete reasons for uniqueness if possible, otherwise consider rephrasing.
We rephrase to: "Together, with VF they form a well-monitored site of glacier-related variables."
- Fig. 1, why show the glacier extent from 2016 and not the present-day? Especially since high-quality whole-glacier imagery is available (Table 1). Also, the 2022/2023 stakes installed and surveyed specifically for the present study should be clearly marked as such and distinguished from the historical archived stakes.
We update Fig. 1, showing the glacier orthoimage from 2022 (newest available). Regarding the glacier extent we now show all available glacier extents, showing the change in the glacier extent, since we also have historical data here. The available extents are listed in the Appendix (see response to major comment 2.1(4)) (1969,1979,1990,1999,2009,2016,2018).

Furthermore, we mark the stakes installed especially for the survey 2022/2023 with a different colour, to distinguish the data from the historical stakes.

updated Fig.1:

[Figure]

- l. 82, maybe it should be mentioned that the stakes were never repositioned at a fixed location, rather they were re-drilled at or close to their last location?

Thanks for the hint. We add in Line 86: "The measurement series are interrupted due to melt out of the stakes. The re-drilling took place close to its last measured position so that the stakes do not describe a fixed position.

- l. 90, "terrestrial polar connection" yields zero results on Google Search - is the method also known by different names? Please check and possibly rephrase or provide a reference

True, change to: "Polar coordinate methods".

- ll. 91-92, the two "previous studies" analyzing velocity at the site appear to be unpublished and unaccessible diploma theses. However, their findings should probably be (1) quickly presented and (2) discussed and compared with the ones of the present study.

As already mentioned in response to major comment 2.2 (3), information contained in these diploma theses are already included in the data. These studies focused on data preparation up to the respective point in time, which is why we would like to mention their valuable work here.

- l. 102, "image pair velocity fields" - I think ITS_LIVE and possibly Millan rather provide annual or biennial velocity composites?

That´s true. We change to: " Figure 2 shows exemplarily the ITS_LIVE (for the period 2017-2018) and Millan data base (annual for 2018) mosaic, …

- l. 113, please provide more details about the TerraSAR-X data and methods used - at least the following: (1) acquisition dates and pairs tested, (2) software used for feature tracking, (3) tracking window sizes used, (4) any post-processing and filtering steps

We provide more information and therefore change line 113-118 to:
Since the user-ready products do not provide usable results, we tested the suitability of high-resolution TerraSAR-X stripmap imagery (~2 m spatial resolution) for obtaining glacier surface velocity fields. Therefore, we applied feature tracking using various tracking window sizes (32x32, 64x64, 128x128, 256x256) and temporal baselines ranging between 11 days and up to about 2 years to pairwise co-registered images. The SAR processing was carried out using GAMMA Remote Sensing Software. Dates and orbit information of the employed acquisitions are summarized in Appendix C. All possible image pair combinations were tested using an automated processing pipeline (e.g. Seehaus et al., 2015, 2018), including an  filter algorithm based on a comparison of the magnitude and the alignment of the displacement vector with surrounding values to remove the unreasonable displacement estimates (Burgess et al., 2012). Coherence tracking or InSAR-based displacement measurements were not feasible to carry out at VF, because the InSAR coherence was not maintained between subsequent acquisitions, which we attribute mainly to the pronounced surface lowering rates in summer and snow accumulation in winter.
Appendix C:

| date | orbit dir. (decending/accending) | rel. orbit number | strip number |
|---|---|---|---|
| 2011-07-03 | D | 78 | 6 |
| 2011-07-14 | D | 78 | 6 |
| 2011-08-08 | A | 131 | 10 |
| 2013-07-01 | A | 131 | 10 |
| 2014-06-07 | A | 131 | 10 |
| 2017-07-11 | D | 78 | 6 |
| 2017-08-13 | D | 78 | 6 |
| 2017-08-24 | D | 78 | 6 |
| 2018-07-31 | D | 78 | 6 |
| 2018-08-22 | D | 78 | 6 |
| 2019-07-29 | D | 78 | 6 |
| 2019-08-31 | D | 78 | 6 |
| 2020-01-02 | A | 131 | 7 |
| 2020-01-30 | A | 55 | 1 |
| 2020-01-30 | A | 55 | 1 |
| 2020-02-21 | A | 55 | 1 |
| 2020-02-21 | A | 55 | 1 |
| 2020-02-26 | A | 131 | 7 |
| 2020-05-02 | A | 131 | 7 |
| 2020-05-08 | A | 55 | 1 |
| 2020-05-08 | A | 55 | 1 |
| 2020-07-04 | D | 78 | 6 |
| 2020-07-26 | D | 78 | 6 |
| 2020-09-11 | A | 131 | 7 |
| 2020-09-17 | A | 55 | 1 |
| 2020-09-17 | A | 55 | 1 |
| 2020-11-11 | A | 55 | 1 |
| 2020-11-11 | A | 55 | 1 |
| 2020-12-25 | A | 55 | 1 |
| 2020-12-25 | A | 55 | 1 |

| date | orbit dir. (decending/accendi | rel. orbit number | strip number |
|---|---|---|---|
| 2021-01-27 | A | 55 | 1 |
| 2021-01-27 | A | 55 | 1 |
| 2021-02-01 | A | 131 | 7 |
| 2021-02-07 | A | 55 | 1 |
| 2021-02-07 | A | 55 | 1 |
| 2021-02-23 | A | 131 | 7 |
| 2021-03-25 | D | 78 | 5 |
| 2021-03-28 | A | 131 | 7 |
| 2021-04-08 | A | 131 | 7 |
| 2021-04-30 | A | 131 | 7 |
| 2021-05-11 | A | 131 | 7 |
| 2021-06-13 | A | 131 | 7 |
| 2021-06-24 | A | 131 | 7 |
| 2021-07-05 | A | 131 | 7 |
| 2021-08-18 | A | 131 | 7 |
| 2021-08-26 | D | 78 | 6 |
| 2021-09-09 | A | 131 | 7 |
| 2021-09-17 | D | 78 | 6 |
| 2021-10-12 | A | 131 | 7 |
| 2021-11-14 | A | 131 | 7 |
| 2021-11-25 | A | 131 | 7 |
| 2022-03-04 | A | 131 | 7 |
| 2022-03-26 | A | 131 | 7 |
| 2022-04-06 | A | 131 | 7 |
| 2022-05-09 | A | 131 | 7 |
| 2022-05-20 | A | 131 | 7 |
| 2022-05-31 | A | 131 | 7 |
| 2022-06-11 | A | 131 | 7 |
| 2022-07-03 | A | 131 | 7 |
| 2022-07-25 | A | 131 | 7 |
| 2022-08-02 | D | 78 | 6 |
| 2022-09-18 | A | 131 | 7 |
| 2022-09-29 | A | 131 | 7 |
| 2022-10-21 | A | 131 | 7 |

- Table 1, please provide details about the "UAV" and the "Optical airborne photogrammetry"
- which platform, camera, flight altitude, spatial resolution?

Information on spatial resolution is added. Details on the platform, camera and flight altitude used in optical aerial photogrammetry are only partially known. Where available, the relevant literature is cited for more information. The UAV data was obtained through manual flights, meaning that no uniform flight altitude was used, as it varies.

updated Table 1:

| Name of dataset | Sensor/Instrument | Date | Res. [m] | Covered area | Data provider |
|---|---|---|---|---|---|
| Airborne 2020 | Optical airborne photogrammetry | 2020-09-08 | 0.2 | Entire glacier | © Land Tirol t i r i s 2020 |
| Airborne 2018 | Optical airborne photogrammetry | 2018-09-16 | 0.2 | Entire glacier | Geissler (2021), Withheld for legal reasons |
| Airborne 2021 | Optical airborne photogrammetry | 2021-09-25 | 0.2 | Entire glacier | Withheld for legal reasons |
| Airborne 2022 | Optical airborne photogrammetry | 2022-08-23 | 0.2 | Entire glacier | Withheld for legal reasons |
| UAV 07/2022 | DJI UAV with optical RGB camera | 2022-07-10 | 0.04 | Parts of Taschach and Brochkogel | 10.5281/zenodo.17590764 |
| UAV 09/2022 | DJI UAV with optical RGB camera | 2022-09-21 | 0.04 | Parts of Taschach and Brochkogel | 10.5281/zenodo.17590764 |
| Stake network (since 1966) | Total-station polar method / GNSS (annual surveys in Sept) | 1966–present (annual) | — | Entire glacier, mainly ablation area | doi.org/10.1594/PANGAEA.982940 |
| Stakes at crevasses 07/2022 | GNSS | 2022-07-10 | — | Parts of Taschach and Brochkogel | doi.org/10.1594/PANGAEA.982940 |
| Stakes at crevasses 08/2022 | GNSS | 2022-08-04 | — | Parts of Taschach and Brochkogel | doi.org/10.1594/PANGAEA.982940 |
| Stakes at crevasses 09/2022 | GNSS | 2022-09-21 | — | Parts of Taschach and Brochkogel | doi.org/10.1594/PANGAEA.982940 |
| Stakes at crevasses 08/2023 | GNSS | 2023-08-08 | — | Parts of Taschach and Brochkogel | doi.org/10.1594/PANGAEA.982940 |

- l. 130, please provide more details here already about this seasonal correction - is it a single multiplication factor? How is it calculated? This information cannot be postponed to the second half of the manuscript.

We add to the manuscript (as later also suggested): " At VF, summer velocities are approximately 30% higher than the annual average. This variation was determined from stake measurements taken at different times (detailed information see chapter 4.2 seasonal variation). However, we consider the derived seasonality to be representative for the entire VF and assume its applicability to other years. The calculated seasonal variation is used to standardize displacement observations recorded at different times, converting them into annual velocity values. A seasonal increase of 30% relative to the annual velocity is assumed for the summer months (July through September)."

- l. 150, first introduce Fig. 3 and what is displayed there, then add specification such as which stakes are included

Done, changed to: "For this purpose, the stake network data, which has been measured annually at VF since 1966, is analyzed. Figure 3 displays the stake positions during the years. The focus lies on evaluating the temporal evolution of the ice movement. Therefore, only time series of at least six consecutive years are considered in the Figure. The data clearly demonstrate …"

- l. 157, the modeling of velocities should be described in detail in the Data and Methods, not just passingly in the Results. In particular, it should be explained (including formulas): (1) Where do the values for local ice thickness come from? (2) How are thickness and surface slope calculated for each stake to evolve for each year? The resulting ice velocity is a high power of both variables, thus it is highly sensitive to their precise values and variations.

Information has been added, for a more detailed description see response to major comment 2.1.

- l. 163, strictly speaking, in the modeled values, we see the sensitivity of velocity to ice thickness, not directly to mass balance.

That´s true, changed to: " A strong sensitivity of velocity to mass balance (shown in Fig. 4) is evident in the measured values, while the modelled values provide the same dynamic trends as the measured. Offsets occur, …"

- l. 165, what is "the temporally fixed choice of flow parameters"? This has not been described before, it has to be fully explained in the Methods subsection about the modeling.

Thanks for the hint. See response to major comment 2.1 (3).

- Fig. 4, this figure is very interesting. However, the "average elevation" of each stake is possibly not the most informative value here, since all elevations are clustered within less than 100 m altitude. If possible, provide also the maximum and minimum elevation of each stake (supposedly corresponding to the earliest and most recent observation dates)

Done.

- Fig. 5, to better highlight interannual changes of ice velocity, it would likely be more informative to display stake velocities not all together as absolute values, but rather as anomalies (additive or multiplicative) compared to the long-term mean at each stake

That's true, we add a subfigure to Fig. 5, showing the anomalies.

[Figure]

- l. 168, no data above 3000 m a.s.l. has been introduced in the text before Fig. 6 - please give a quick introduction to those stakes, are they all stakes with < 6 consecutive years of data? Are they stakes in the accumulation area?

We do not understand this point of the reviewer. Figure 6 shows the whole dataset of measured velocities; no differentiation was made e.g. on length of time series. We updated the Figure caption accordingly.

Regarding the point "no data above 3000m": We basically have velocity measurements in the ablation zone, since the stake network was designed to measure ablation rather than the velocity pattern of VF (see lines 82-84 of current version of manuscript). To clarify, we added in line 84 of the current version of manuscript: "Note that the stake network was adapted over the years to capture the ablation area which changed over the years due to the propagation of ablation to higher elevation in response to global warming. Therefore, the elevation range captured by the stake network varies.

- l. 172, these are all methodological details that belong to the Methods section, as they are necessary to understand most of the results presented so far. Also, more details are needed on the calculation of ice thickness change: was it computed only from local mass balance,

fully neglecting ice advection? This would introduce a major, systematic, elevation-dependent bias in the ablation area (too fast thinning)

We agree and move these methodological details to the Methods section. Details on the calculation of ice thickness are explained in response to major comment 2.1. Both, mass balance and mass transport are taken into account in the surface models. We interpolate between two surface models. The mass balance is only used to scale the geometric change. The geodetic height change includes ice advection, so at least the average ice transport is taken into account.

- Fig. 6, this interesting plot is hard to read, especially since all point measurements from all stakes are shown without distinction. At this stage, the reader does not know how many stakes are visualized on this plot, and how many stakes exist at each given point in time. Thus, it is not clear what is the time evolution here, apart from a general trend of "slowdown at all altitudes". It could make more sense to connect the points of single stakes, possibly aggregating in multi-annual intervals (even decadal aggregation) and/or excluding stakes with very few years of observation, in order to reduce the complexity of the data shown.

We are not sure whether we have correctly understood the suggestion regarding Fig. 6. Representing stakes with a long observation period and connecting the points of single stakes would lead to a Figure close to Fig. 4 of the original manuscript. We agree that it is not possible to distinguish between the individual stakes in Fig. 6 in the current way, but we intent to show that the glacier velocity slow down occurs in all height levels. Therefore, we keep this Figure in the manuscript.

- Fig. 7, the thickness maps exhibit obvious major interpolation/processing artifacts, which would be strongly reflected in any calculated ice velocity. The thickness data need to be made available and/or re-examined in the light of these artifacts, if any conclusion about changes in glacier geometry is to be drawn. Also, if possible, please show simultaneous extent and thickness of the glacier rather than inconsistent dates; the 2016 extent is already in Fig. 1. Finally, the arrows indicate the flow direction, how was it determined? Please provide methodological details in the relevant section.

That is true, thanks for the hint. We re-examined these artifacts (see new figure 7 in this comment).

In addition, the publication of the DGMs (see response to major comment 2.1(4)) allows the calculation of ice thickness data to be traced.

In Figure 7, we show the average stake velocities for three different periods (1970–1980, 1990–2000, and 2010–2020). In order to represent the average ice thickness for each epoch, we decided to show the ice thickness for the respective average year (1975, 1995, 2015). Unfortunately, we do not have glacier extent data for these exact years, therefore the ice thickness is clipped to the last known glacier extent, as it is described in more detail in the response to major comment 2.1(1). To avoid misunderstandings, and since the clipping is already described, the glacier extent is removed from Fig. 7.

The arrows indicate the average direction of movement of the respective stake. This information is added to the figure caption: "The arrows indicate the determined flow direction, which is calculated from the mean angle of movement of the respective stake."

updated Fig.7:

[Figure]

a) annual stake velocity between 1970 and 1980    b) annual stake velocity between 1990 and 2000

c) annual stake velocity between 2010 and 2020

ice thickness (m)

velocity (m/yr)

- l. 184, here is a description of a methodological choice and would fit very well around l. 130 if it is the same seasonality correction that is described here.

Done, thanks.

- Fig. 8, what is the standard deviation of this 30 % summer speed-up? If I understand correctly, the monthly (in summer) and annual velocities are available for each of the 8 + 4 stakes mentioned here, thus it would be quite important to show how much these stakes deviate from the 30 % estimate (and thus, how uncertain the estimate is) - especially if the correction factor is to be considered "representative and applicable" anywhere.

Velocity data used to calculate seasonal variations is available at PANGAEA. An additional table is added to explicitly show the individual vales and the calculated seasonal variation.:

**Table 2.** Exact values of the seasonal variation of glacier surface velocity for Taschach (8 stakes) and Brochkogel area (4 stakes).

| Area | Number | $v_{Jul-Aug}$ Juli 22–Aug 22 [m yr$^{-1}$] | $v_{Aug-Sep}$ Aug 22–Sep 22 [m yr$^{-1}$] | $v_{annual}$ Aug 22–Aug23 [m yr$^{-1}$] | $\frac{v_{Jul-Aug}}{v_{annual}}$ seas. var. | $\frac{v_{Aug-Sep}}{v_{annual}}$ seas. var. |
|---|---|---|---|---|---|---|
| Brochkogel | 1 | 3.37 | 5.05 | 3.14 | 1.07 | 1.61 |
| Brochkogel | 2 | 3.44 | 4.74 | 3.03 | 1.14 | 1.56 |
| Brochkogel | 3 | 4.52 | 4.16 | 3.22 | 1.40 | 1.29 |
| Brochkogel | 4 | 4.62 | 3.00 | no measurement | | |
| **Brochkogel** | **mean** | **3.99** | **4.24** | **3.13** | **1.20** | **1.49** |
| | | | | | | |
| Taschach | 1 | 4.54 | 2.21 | 2.38 | 1.91 | 0.93 |
| Taschach | 2 | 3.78 | 2.39 | 2.27 | 1.67 | 1.05 |
| Taschach | 3 | 3.57 | 3.85 | 3.09 | 1.16 | 1.25 |
| Taschach | 4 | 4.12 | 3.97 | 2.17 | 1.90 | 1.83 |
| Taschach | 5 | 1.27 | 1.52 | 1.74 | 0.73 | 0.87 |
| Taschach | 6 | 1.82 | 3.84 | 2.11 | 0.86 | 1.82 |
| Taschach | 7 | 2.24 | 3.49 | 2.17 | 1.03 | 1.61 |
| Taschach | 8 | 2.08 | 1.95 | 1.77 | 1.18 | 1.10 |
| **Taschach** | **mean** | **2.93** | **2.90** | **2.21** | **1.30** | **1.31** |

We add to the manuscript:

in current line 221:

On average, seasonal variation is 1.30 (130%), with a standard deviation across all observations of 0.37 (37%).

and in current line 311:

At Vernagtferner, summer velocities are approximately 30% higher than the annual mean, with a standard deviation across all observations of 37%. The relatively high uncertainty results most likely from the fact that the absolute measured values over a month (July-Aug or Aug-Sep) have maximum values of about 50 cm. However, these values have a relatively high measurement uncertainty of +/-14cm (see chapter Measurement Uncertainty).

- l. 218, a claimed 3 cm measurement error corresponds to a good-quality dGNSS / RTK survey, whose method (instruments and protocol) should be described in the Methods section.

We will name the method in Table 1. In our opinion, GNSS measurement is a state-of-the-art method. Therefore, we do not describe this method explicitly in the "Methods" section and do not submit any protocols.

This was a classic static GNSS measurement using a multi-frequency GNSS receiver, with our own base station in operation. The post-processing evaluation, which we had to rely on due to a lack of data connection for RTK processing, consistently delivered solutions with fixed ambiguities. This resulted in an accuracy of approximately 3 cm. We are adding this information to Table 1.

- l. 222, the data resolution should be introduced in the presentation of the UAV / aerial data.

Done, see response to major comment 2.1 (4).

- l. 229, the statistical basis of this calculation of overall uncertainty is unclear: the mean uncertainty is most definitely not the average of individual uncertainties. See e.g. [12], [13]. Another possibility would be to use leave-one-out validation of each available measurement.

That´s true. We change the uncertainty analysis, a new version can be found in the response to major comment 2.6

- l. 233, the dataset contains maps of ice velocity and of surface slope, it should be easy to calculate proper metrics (such as a correlation coefficient) of the agreement between surface aspect and flow direction, rather than a qualitative "overall alignment of the calculated flow directions with the glacier topography".

That´s true. We calculate the angle deviation of the flow direction of the topography to the flow direction of the velocity in the median at 43 degrees. Larger angular deviations may occur, especially at lower magnitudes.

- l. 235, there exist published methods to estimate interpolation error in areas of heterogeneous point data coverage. See for example [14].

This must be a misunderstanding; we don't interpret here the quality of the interpolation. We just say the ice flow map is most trustworthy in areas of high data density. We slightly rewrite the sentence for clarity: "Despite the estimated interpolation error, the velocity map becomes less reliable in regions with limited data availability. In zones where observational data is lacking, there is insufficient information, making the reliability of the velocity map for understanding ice flow dynamics in these areas questionable."

However, we think the reviewer comment also belongs to lines 231-234 where we estimate the interpolation error. More information, including testing of the method described in the literature suggestion [14], can be found in the response to major comment 2.6.

- l. 253, is this "wide range of notable differences between the summer and winter seasons" shown anywhere? So far the Authors have presented only three spatially-aggregated, monthly data points from a single summer season (Fig. 8).

As shown in Table 2 above, the datapoints are represented now. We agree, the description as "wide range of notable differences between summer and winter seasons" is not sufficiently justified based on the initial data, we change it to: "with a notable difference between summer and winter seasons".

Information on the restrictions regarding the summer year and the spatial extent can already be found in lines 256-267.

- l. 315, this conclusion can only be taken if proper state-of-the-art methods are tested and the results are shown, the well-established methods validated by several studies (including on slow glaciers) cannot be so quickly dismissed.

We agree. We don´t want to rule out the possibility that there could be a successful tracking of the features. At this point, we would like to draw more attention to the overlap caused by ablation. Even if a modern method enables feature tracking, the challenges associated with high ablation remain. We would like to clarify this statement as follows:

"With the general tendency of strong negative mass balances and the associated possibility of high ablation, changes in surface features may be caused in parts by ablation rather than ice dynamics. This may also make it difficult to detect ice dynamics in other Alpine glaciers under climate change.

- l. 323, I would caution against using a map with manually tracked features as benchmark for validating other datasets, since the computed quality of such datasets would then inherit the reproducibility issues of the manually-compiled map.

We agree and rewrite: …. allow a detailed modeling of glaciological processes as well as plausibility assessments of future remote-sensing results."

- l. 329, data availability: most of the datasets mentioned and used within the study are actually missing, including the ice thickness data, any digital elevation models, and the UAV and airborne data. Only the historical stake data are provided, the other assets are rather results from the study such as the manually tracked displacements and the five-year velocity map. Unless there are legal restrictions, for both review and reproducibility purposes it is important to provide access to the actual data (not just the results) used in the study.

We will publish all data that is not subject to legal restrictions. An overview can be found in the new created table, see response to major comment 2.1 (4).

Additional Literature used in the response:

**Burgess**, E.W.; Forster, R.R.; Larsen, C.F.; Braun, M. Surge dynamics on Bering Glacier, Alaska, in 2008–2011. Cryosphere 2012, 6, 1251–1262. doi:10.5194/tc-6-1251-2012, **2012**

**Gindraux**, S.: The potential of UAV photogrammetry for hydro-glaciological forecasts, Mitteilungen der Versuchsanstalt für Wasserbau, Hydrologie und Glaziologie (VAW), ETH Zürich, No. 252, Zürich, **2019**

**Greve**, R. and Blatter, H.: Dynamics of ice sheets and glaciers, Springer, Dordrecht, ISBN-13 978-3642034145, ISBN-10 3642034144, **2009**

**Hutter**, K.: Theoretical glaciology: material science of ice and the mechanics of glaciers and ice sheets, D. Reidel Publishing Co., Dordrecht/Terra Scientific Publishing Co., Tokyo, ISBN-10 9401511691, ISBN-13 978-9401511698, **1983**

**Mattea,** Enrico; Berthier, Etienne; Dehecq, Amaury; Bolch, Tobias; Bhattacharya, Atanu; Ghuffar, Sajid; Barandun, Martina; Hoelzle, Martin. Five decades of Abramov glacier dynamics reconstructed with multi-sensor optical remote sensing. The Cryosphere, 219-247. doi:10.5194/tc-19-219-2025, **2025**

**Paterson,** W.S.B.,1994. The Physics of glaciers, Third edition,doi: 10.1016/C2009-0-14802-X, **1994**

**Seehaus**, T., Marinsek, S., Helm, V., Skvarca, P., Braun, M., 2015. *Changes in ice dynamics, elevation and mass discharge of Dinsmoor–Bombardier–Edgeworth glacier system, Antarctic Peninsula.* Earth and Planetary Science Letters 427, 125–135. https://doi.org/10.1016/j.epsl.2015.06.047, **2015**

**Seehaus**, T., Cook, A.J., Silva, A.B., Braun, M., 2018. *Changes in glacier dynamics in the northern Antarctic Peninsula since 1985.* The Cryosphere 12, 577–594. https://doi.org/10.5194/tc-12-577-2018, **2018**

**Vincent**, C.; Vallon, M.; Reynaud, L.; Le Meur, E. Dynamic behaviour analysis of glacier de Saint Sorlin, France, from 40 years of observations, 1957–97, Journal of Glaciology,499-506. doi: 10.3189/172756500781833052, **2000**

**Vincent**, C.; Soruco, A.; Six, D.; Le Meur, E. Glacier thickening and decay analysis from 50 years of glaciological observations performed on Glacier d'Argentière, Mont Blanc area, France, Annals of Glaciology, 73-79, doi: 10.3189/172756409787769500, **2009**

**Vincent**, Christian; MOREAU, L. U.C., 2016. Sliding velocity fluctuations and subglacial hydrology over the last two decades on Argentière glacier, Mont Blanc area. Journal of Glaciology 62, 805-8015. doi: 10.1017/jog.2016.35, **2016**